# Attention De-sparsification Matters: Inducing Diversity in Digital Pathology Representation Learning

## Abstract

In this work, we develop *DiRL*, a **D**iversity-**i**nducing **R**epresentation **L**earning technique for histopathology image analysis. SSL techniques, such as contrastive and non-contrastive approaches, have been shown to learn rich and effective representations without any human supervision. Lately, computational pathology has also benefited from the resounding success of SSL. In this work, we develop a novel prior-guided pre-training strategy based on SSL to enhance representation learning in digital pathology. Our analysis of vanilla SSL-pretrained models' attention distribution reveals an insightful observation: *sparsity in attention*, i.e, models tends to localize most of their attention to some prominent patterns in the image. Although attention sparsity can be beneficial in natural images due to these prominent patterns being the object of interest itself, this can be sub-optimal in digital pathology; this is because, unlike natural images, digital pathology scans are not object-centric, but rather a complex phenotype of various spatially intermixed biological components. Inadequate diversification of attention in these complex images could result in crucial information loss. To address this, we first leverage cell segmentation to densely extract multiple histopathology-specific representations. We then propose a prior-guided dense pretext task for SSL, designed to match the multiple corresponding representations between the views. Through this, the model learns to attend to various components more closely and evenly, thus inducing adequate diversification in attention for capturing context rich representations. Through quantitative and qualitative analysis on multiple slide-level tasks across cancer types, and patch-level classification tasks, we demonstrate the efficacy of our method and observe that the attention is more globally distributed. Specifically, we obtain a relative improvement in accuracy of up to 6.9% in slide-level and 2% in patch level classification tasks (corresponding AUC improvement up to 7.9% and 0.7%, respectively) over a baseline SSL model.

## 1 Introduction

Computational pathology is a rapidly emerging field that aims at analyzing high resolution images of biopsied or resected tissue samples. Advancements in computer vision and deep learning has enabled learning of the rich phenotypic information from whole slides images (WSIs) to understand mechanisms contributing to disease progression and patient outcomes. Acquiring crop-level localized annotations for WSIs is expensive and often not feasible; only slide-level pathologist labels are usually available. In such a scenario, weak supervision is a commonly utilized strategy, where crops are *embedded into representations* in the first stage, followed by considering these WSI-crops' representation as a bag for multiple instance learning (MIL). Now the question remains, *how do we learn a model to effectively encode the crops into rich representations?* Traditionally, ImageNet (Krizhevsky et al., 2017) pre-trained neural networks are utilized to extract the representations (Lu et al., 2021b; Lerousseau et al., 2021; Shao et al., 2021). However ImageNet and pathology datasets are composed of different semantics; while the former contains object-centric natural images, the later consists of images with spatially distributed biological components such as cells, glands, stroma, etc. Therefore, to learn domain-specific features of WSI-crops in the absence of localized annotations, various self-supervised learning (SSL) techniques are recently gaining traction (Ciga et al., 2022; Stacke

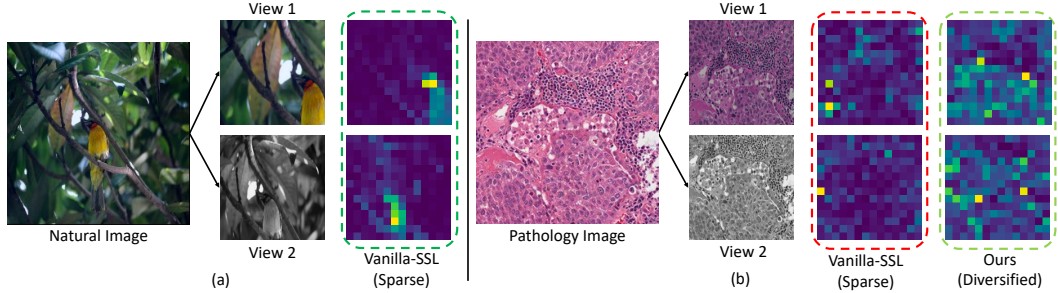

Figure 1: **Diversification of attention for encoding dense information in digital pathology.** View 1 and View 2 are two augmented views of the input image. a) Illustration of attention map from a model pre-trained on ImageNet using vanilla SSL. b) Attention map of model pre-trained on histopathology dataset with vanilla SSL, and with our proposed pre-training strategy. In both natural imaging and digital pathology, vanilla SSL pre-training creates *sparse* attention maps, i.e., it attends largely to only some prominent patterns. Although attention sparsification can be beneficial in natural image tasks such as object classification, this could be sub-optimal for encoding representations in digital pathology as it leads to loss of important contextual information. Through a more diversified attention mechanism, *DiRL* encodes dense information critical to non object-centric tasks.

et al., 2021; Boyd et al., 2021). SSL models pre-trained on histopathology datasets have been shown to be effective in downstream classification tasks when compared to those trained on ImageNet.

To further analyze the role of SSL in computational pathology, we pre-trained a vision transformer (Dosovitskiy et al., 2020) on various WSI datasets using vanilla SSL (Caron et al., 2021). In-depth analysis of the pre-trained models' attention maps on WSI-crops led us to a striking observation: **sparsity in attention maps**. The model tends to localize most of its attention to a small fraction of regions, leading to sub-optimal representation learning. To further validate our observation, we visualized the attention maps of a self-supervised ImageNet pre-trained model on natural images (see Fig. 1). Similar observations led us to conclude that this is a property of SSL rather than of data. We believe that sparsity in attention might potentially benefit the performance in some natural imaging tasks such as object classification. This stems from the fact that during SSL, the model is tasked to match the two views, optimizing which leads the model to focus on the prominent patterns. For example, in Fig. 1(a), for an object-centric ImageNet example, since the prominent pattern is the object (eg. bird) itself (Yun et al., 2022), the model tends to center its attention towards the object, thus benefiting numerous downstream applications (for eg., bird classification). In contrast, WSI-crops are *not object-centric*, rather they constitute a *spatial distribution of complex structures such as cells, glands, their clusters and organizations, etc*, see Fig. 1(b). Encoding this dense information available into a holistic representation demands the model to focus more diversely to various histopathology primitives and not just to specific ones. Conversely, the vanilla SSL model pre-trained on histopathology only sparsely attends to the important regions (Fig. 1(b)), i.e., there is inadequate diversity in attention. We *hypothesize that this sparsely attending model could result in encoding sub-optimal representations, as fine-grained context-rich details are often ignored*.

To address this issue of inadequate attention diversity, we propose *DiRL*, a diversity-inducing pre-training technique, tailored to enhance representation learning in digital pathology. Each WSI-crop consists of *two regions: cellular regions (one containing cells) and non-cellular regions (containing no cells)*. We leverage an off-the-shelf cell segmentation pipeline to identify these regions. This domain-specific knowledge is then utilized to extract **region-level representations** separately for the cellular and non-cellular regions. We further propose to encode the inter- and intra-spatial interplay of two regions. This biologically-inspired step (Saltz et al., 2018; Fassler et al., 2022) is achieved through a transformer-based disentangle block to encode the self-interaction within the regions, and cross-interaction between both the regions, termed as **disentangled representations**. In contrast to vanilla SSL frameworks that leverage one image-level representation for a WSI-crop, our prior-guided representation learning framework leverages histology-specific domain knowledge to densely extract a set of region-level and disentangled representations. We then task our framework to match all the corresponding representations between the views. We hypothesize that *optimizing this dense matching objective between the views would encourage the model to diversify its attention to various regions; matching assorted representations would then enforce the model to explore diverse*

*image-regions relevant for each such representations.* We validate this hypothesis through consistent improvements in performance on multiple downstream tasks such as slide-level and patch-level classifications. Our qualitative analysis on attention distribution of the pre-trained models reveals that our *DiRL* framework can effectively de-sparsify attention, thereby learning global context-rich representations, unlike existing methods.

To summarize our main contributions, we:

- Demonstrate that attention sparsification in self-supervised learning may lead to learning sub-optimal representations in digital pathology classification tasks.

- Design a novel domain-aware pretext task to de-sparsify attention maps and achieve enhanced representations for digital pathology.

- Demonstrate the efficacy of our *DiRL* through slide-level and patch-level classification tasks on four WSI datasets and two patch datasets.

## 2  RELATED WORK

In this section, we briefly discuss vision transformers, SSL and its dense counterpart, and their application in computational pathology.

**Vision transformers.**  Inspired by the success of self-attention modules in language models (Vaswani et al., 2017), vision transformers (ViTs) Dosovitskiy et al. (2020); Liu et al. (2021); Touvron et al. (2021); Xu et al. (2022); Ali et al. (2021); Tu et al. (2022) have been have been proposed to exploit non-local spatial dependencies in the imaging domain. Recent studies Wang et al. (2021b); Chen & Krishnan (2022); Chen et al. (2022); Stegmüller et al. (2022); Gao et al. (2021); Chen et al. (2021) have demonstrated the promise of transformer-based architectures in modeling histopathology imaging for cancer diagnosis and prognosis. However, to the best of our knowledge, no existing work has leveraged the flexibility of attention mechanism in transformers to instill the biology-relevant domain knowledge into vision transformers. For example, interaction between concepts/primitives such as tumor nuclei and stroma or between lymphocytic cells plays an important role in disease pathophysiology and treatment outcome. Our proposed method takes a step in this direction through a domain-driven pretext task.

**Image-level SSL** aims at learning visual representations through different pretext tasks. Contrastive and non-contrastive methods such as Chen et al. (2020); He et al. (2020); Caron et al. (2021), have shown tremendous potential in learning robust and rich representation in natural imaging. Building upon them, studies such as Ciga et al. (2022); Stacke et al. (2021); Li et al. (2021a); Kapse et al. (2022); Boyd et al. (2021); Chen & Krishnan (2022); Kurian et al. (2022) have explored SSL pretraining in histopathlogy image analysis.

**Region-level SSL** aims to further boost information encoding through dense pre-training techniques such as Li et al. (2021b); Yun et al. (2022); Wang et al. (2021a). These techniques impose additional constraints to match 1) region-level correspondences across the two views of the data or 2) neighbor-level intra-view correspondences within the data. Studies such as Wen et al. (2022); Yang et al. (2022); Hénaff et al. (2021) have explored utilizing segmentation-based or clustering-based regions in self-supervision to enhance representation learning. However, the goal of these studies is to mainly improve the transfer performance for dense-prediction tasks such as *object detection* and *segmentation*. In contrast, we tailored a dense pre-training strategy in histopathology to enforce the model to focus on diverse-regions thus diversifying model's attention. This diversified attention encourages the model in effectively encoding the complex information about various histology components, thereby augmenting *classification* performance.

## 3  PROPOSED METHOD

In this section, we first describe a naïve vision transformer framework for Whole Slide Images (WSIs). This is followed by explaining how cell segmentation can be used as a prior in pre-training for WSIs. Next, we present the extraction of region-specific representations using our proposed cell-back pooling and disentangle block. Finally, we present *DiRL*, our diversity-inducing pre-training technique, to learn discriminative features for WSI patches which are subsequently leveraged by a multiple instance learning (MIL) framework for downstream classification tasks. An overview of the proposed architecture is shown in Fig. 2(a).

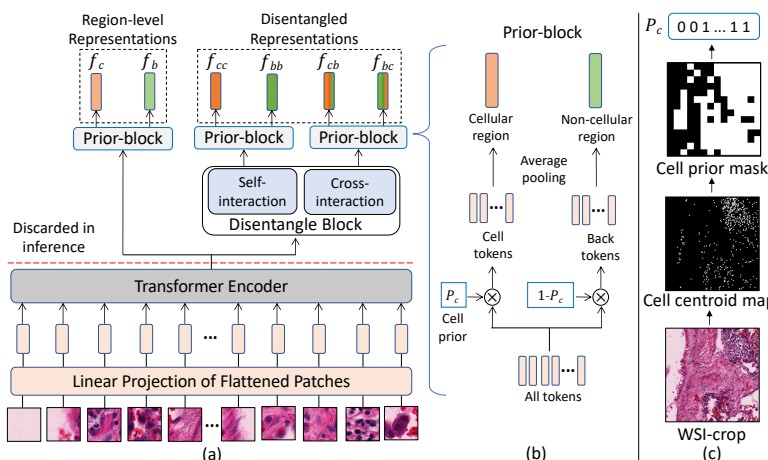

Figure 2: **Overview of the proposed *DiRL* framework:** a) A WSI-crop is patchified and fed into a linear projection layer followed by a transformer encoder. The output is fed to a Prior-block and to a Disentangle block. The Prior-block pools region-level representations separately for cellular and non-cellular regions. The Disentangle block encodes spatial interplay between the two regions followed by prior-block to extract region-level disentangled representations. b) Cell priors $P_c$ and $1-P_c$ pool the tokens associated with the cellular and the non-cellular regions, respectively, followed by average pooling to extract region-level features. c) Cell segmentation from WSI-crop followed by extraction of cell centroid map. Cell prior mask is generated by discretizing the cell centroid map into patches. A Cell prior vector $P_c$ is then produced from the cell prior mask. After pre-training, modules above the red dashed line are discarded in the inference stage.

**Preliminaries.** For an understanding of the primary components in a transformer such as MSA (Multi-head Self-Attention), LN (Layer Normalization), and MLP (Multi-Layer Perceptron), we refer the readers to Vaswani et al. (2017).

### 3.1 VISION TRANSFORMER FOR WSI

From each WSI, $\mathcal{W}$, $w_1, w_2, \ldots w_N$ crops are extracted, where $N$ is variable for each $\mathcal{W}$. Each $w_i$ is then decomposed into $n$ patches $\mathcal{X} = [X^1, X^2, ..., X^n] \in \mathcal{R}^{n \times p \times p \times 3}$, where $(p, p)$ is the spatial size of each patch. Each patch is transformed into a token using a shared linear projection layer,

$$\mathcal{T}_0 = [X^1\mathbf{E}; X^2\mathbf{E}; ..., X^n\mathbf{E}] \tag{1}$$

where $\mathbf{E}$ are convolutional filters operating on each patch with $d$ number of $p \times p$ size filters, thus extracting a $d$ dimensional feature vector for patch. This is followed by adding 1D learnable position embedding as in Vaswani et al. (2017). The transformer block models the relationship between the tokens using a multi-head self-attention block:

$$\mathcal{T}_l' = \mathcal{T}_{l-1} + \text{MSA}(\text{LN}(\mathcal{T}_{l-1})); \qquad \mathcal{T}_l = \mathcal{T}_l' + \text{MLP}(\text{LN}(\mathcal{T}_l')) \tag{2}$$

where $l$ is index of the $l^{th}$ block of transformer encoder, composed of $L$ stacked transformer blocks. Thus in each block, these tokens interact with each other to learn representations for each $w_i$. The resulting $\mathcal{T}_L$ of dimension $(n, d)$ is average pooled across all the $n$ tokens to compute the image-level representation $f$ of dimension $(1, d)$ as shown in Fig. 3(a).

### 3.2 CELL SEGMENTATION AS DOMAIN PRIOR

Each WSI-crop $w_i$ consists of **two regions**, one containing cells and the other without cells. There has been substantial advancements in deep learning research pertaining to cell segmentation; this stems from the important role of image analysis and machine learning algorithms in visual interpretation of cellular biology (morphology and spatial arrangement) in digitized pathology scans (Lu et al., 2021a; Shaban et al., 2022; Ding et al., 2022). Identifying the cellular and non-cellular regions in $w_i$ can be achieved by exploiting the cell segmentation output as prior via techniques such as Sahasrabudhe et al. (2020); Hou et al. (2020); Vahadane & Sethi (2013). Following cell segmentation, the centroids are extracted to yield the cell centroid map, a binary map ($\mathcal{C}$) of values zeros and ones, with $\mathcal{C}_{i,j} = 1$ if centroid of any cell is present at $(i, j)$ pixel in WSI-crop. We term this as cell prior

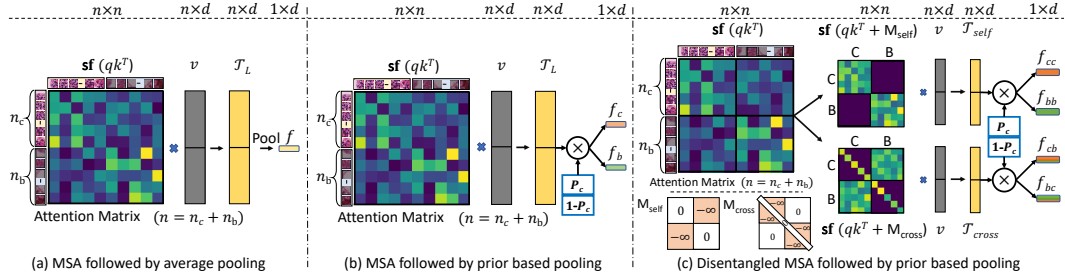

Figure 3: a) Illustration of the $n \times n$ attention matrix from the last layer of the transformer encoder, where $q, k, v$ are projections of tokens in transformer block. Matrix multiplication of attention matrix with value $v$, followed by average pooling across all the tokens generates the representation ($f$) for $w_i$. b) Cell prior $P_c$ is utilized to separately pool cell tokens and back tokens to extract region-level representations. c) Tokens from the $L^{th}$ layer are interacted in the transformer-based disentangle block, forming the attention matrix. Attention masks $M_{self}$ and $M_{cross}$ are added with the attention matrix, generating desired matrices for disentanglement. Note that sf denotes softmax activation. Matrices are then multiplied with $v$ followed by prior-based pooling, thus extracting four representations encoding spatial interplay in $w_i$. For clarity, the cellular and non-cellular region patches ($n_c$ and $n_b$ respectively) are arranged in separate groups.

mask as shown in Fig. 2(c). Since cell segmentation is routinely used in computational pathology, we use off-the-shelf, well established cell segmentation pipelines instead of training a new model. To be coherent with ViT, $\mathcal{C}$ is decomposed into $n$ patches $\mathcal{C} = [C^1, C^2, ..., C^n]$ of size $(p, p)$. Each patch is transformed as follows: $C^i = \mathrm{MaxPool}(C^i)$, i.e., if the patch $C^i$ contains one or more centroids, it becomes one, else it remains zero. Thus, the cell prior mask is downsampled into a binary vector of dimension $(n, 1)$, denoting the presence of cells in each patch of $w_i$. We term this vector as cell prior ($P_c$), which is invoked to extract the region-specific representations for each $w_i$.

### 3.3 PRIOR-BLOCK FOR CELL-BACK POOLING

Following $L$ stacks of transformer blocks, a set of tokens $\mathcal{T}_L \in \mathcal{R}^{n \times d}$ is fed to a prior-block. In this block, the tokens can be categorized into (a) *cell* tokens, implying the tokens whose input patches contain at least a cell and (b) background or *back* tokens, whose input patches do not capture any cells. The cell and back tokens are separately encoded to represent region-level features from the cell prior $P_c$ as follows:

$$f_c = \frac{P_c^T \mathcal{T}_L}{\sum P_c}; \qquad f_b = \frac{(1 - P_c)^T \mathcal{T}_L}{\sum 1 - P_c} \tag{3}$$

$f_c$ is the average pooled representation of all the cell tokens, i.e. representation of the cellular region. $f_b$ is the average pooled representation of all the back tokens, representing non-cellular regions. In the process, *Cell-Back Pooling* is exploited in the prior-block to extract two **region-level representations** as shown in Fig. 3(b).

### 3.4 DISENTANGLE BLOCK

We take a step towards obtaining region-level representation by proposing a transformer block for disentangling the cellular and non-cellular regions. This disentanglement is performed to encode self-interaction in each region and cross-interaction between the two regions. To accomplish this disentanglement, we devise two attention masks, $M_{self}$ and $M_{cross}$, each of dimension $(n \times n)$ as shown in Fig. 3(c). The goal of $M_{self}$ is to only allow token interaction between the same regions, i.e., cell-cell and back-back. In contrast, $M_{cross}$ allows tokens to interact between the different regions (cell-back). The masks $M_{self}$ and $M_{cross}$ are computed as:

$$M_{self}(i,j) = \begin{cases} 0, & \text{if } P_c(i) = P_c(j) \\ -\infty, & \text{otherwise} \end{cases}; \quad M_{cross}(i,j) = \begin{cases} 0, & \text{if } P_c(i) \neq P_c(j) \text{ or } i = j \\ -\infty, & \text{otherwise} \end{cases} \tag{4}$$

where indices $i, j \in \{1, 2, ...n\}$. Recall that in transformers, tokens are projected into three embeddings $q, k, v$ and the output of a MSA block is computed as a weighted sum of the values $v$, where

the weight assigned to each value is determined by a self-attention operation $\text{softmax}(qk^T)$. Unlike standard MSA in $w_i$, where all the tokens from cellular and non-cellular regions interact with each other through self-attention (see Fig. 3(a)), we propose to disentangle the interactions between the regions. $M_{self}$ and $M_{cross}$ are linearly combined with the attention matrix to obtain disentangled self-attention matrices as follows:

$$\text{MSA}_{self} = \text{softmax}(qk^T + M_{self}); \quad \text{MSA}_{cross} = \text{softmax}(qk^T + M_{cross}) \quad (5)$$

Note that attention masks $M_{self}$ and $M_{cross}$ are linearly combined before the $\text{softmax}$ activation to ensure that the sum of each row in the self-attention matrix remains one. The disentangle block operates at the output of the transformer encoder $\mathcal{T}_L$ as:

$$\mathcal{T}'_{self} = \mathcal{T}_L + \text{MSA}_{self}(\text{LN}(\mathcal{T}_L)); \quad \mathcal{T}_{self} = \mathcal{T}'_{self} + \text{MLP}(\text{LN}(\mathcal{T}'_{self})) \quad (6)$$

$$\mathcal{T}'_{cross} = \mathcal{T}_L + \text{MSA}_{cross}(\text{LN}(\mathcal{T}_L)); \quad \mathcal{T}_{cross} = \mathcal{T}'_{cross} + \text{MLP}(\text{LN}(\mathcal{T}'_{cross})) \quad (7)$$

Finally, similar to 3.3, the region-level features are encoded using the cell prior $P_c$ using:

$$f_{cc} = \frac{P_c^T \mathcal{T}_{self}}{\sum P_c}; \quad f_{bb} = \frac{(1 - P_c)^T \mathcal{T}_{self}}{\sum 1 - P_c}; \quad f_{cb} = \frac{P_c^T \mathcal{T}_{cross}}{\sum P_c}; \quad f_{bc} = \frac{(1 - P_c)^T \mathcal{T}_{cross}}{\sum 1 - P_c} \quad (8)$$

Thus, the prior-based pooling on $\mathcal{T}_{self}$ and $\mathcal{T}_{cross}$ results in four **disentangled representations** $f_{cc}$, $f_{bb}$, $f_{cb}$, and $f_{bc}$, encoding the spatial interplay between the cellular and non-cellular regions. Thus, for each WSI-crop $w_i$ *we encode six representations: two region-level representations using cell-back pooling, and four disentangled representations using disentangle block.* Our prior-guided pre-training framework operates on these six representations to pre-train the model.

### 3.5 DIVERSITY-INDUCING PRE-TRAINING FOR WSI

In this section, we formulate our diversity-inducing representation learning (*DiRL*) using a widely used SSL framework for pre-training on histopathology data: DINO (Caron et al., 2021). However, in practice, our pre-training technique can be integrated with any pairwise SSL framework (Li et al., 2022), as demonstrated in Appendix A.3. DINO consists of a student and teacher branch, where the teacher is a momentum updated version of the student, thus both having same architecture (models). Different views of the input image are fed to both the branches to encode them into image-level representations. A projection head is applied on top of these representations with $\text{softmax}$ activation. SSL is performed by matching the student's output with the teacher's probability distribution through cross-entropy loss, $\mathcal{L}^{CE}$. In contrast to vanilla DINO, *DiRL* yields six feature vectors from each branch (see Fig. 2). Therefore, the loss function is modified as:

$$\mathcal{L}^{CE} = \lambda_1 \times (\mathcal{L}_c^{CE} + \mathcal{L}_b^{CE}) + \lambda_2 \times (\mathcal{L}_{cc}^{CE} + \mathcal{L}_{bb}^{CE} + \mathcal{L}_{cb}^{CE} + \mathcal{L}_{bc}^{CE}) \quad (9)$$

where $\mathcal{L}_c^{CE}$ is the cross-entropy loss between projection of representation $f_c$ of student and that of teacher branch. Likewise, the projected distribution of all other corresponding representations from student and teacher are matched. This linear combination of losses encourages the framework to perform a dense matching of the region-level and disentangled representations of the augmented views. Consequently, the dense matching promotes the model to globally diversify the attention map (refer to Fig 1, Fig 12).

We propose another variant of *DiRL*, without the disentangle block, i.e. similarities of only projection distribution of $f_c$ and $f_b$ are maximized between the views. We name this variant as *Cellback*. Following the pre-training, only linear projection layer, position embedding, and the transformer encoder of the teacher are retained. This pre-trained ViT extracts the average pooled feature representation for all $w_i$ belonging to WSI $\mathcal{W}$, to generate feature matrix of dimension $(N, d)$, where $N$ is variable number of WSI-crops for each $\mathcal{W}$. Note that the prior is used *only* at pre-training. Finally, MIL operates over this matrix for WSI slide-level analysis, as discussed next.

### 3.6 MULTIPLE INSTANCE LEARNING FOR SLIDE-LEVEL TASKS

Multiple instance learning (MIL) is widely used method in WSI slide level analysis. We refer the readers to Ilse et al. (2018); Lerousseau et al. (2021); Shao et al. (2021); Lu et al. (2021b) for an overview. We adopted DSMIL (Li et al., 2021a) framework for this work. Following pre-training, the pre-trained model is used to extract features for WSI-crops in $\mathcal{W}$. The MIL model takes these features as input bag, optimizing the model weights through slide-level label supervision.

## 4 EXPERIMENTS AND RESULTS

Our pre-trained models are evaluated for both slide-level and patch-level classification tasks. As a *Baseline*, we pre-trained a vision transformer with DINO (Caron et al., 2021), a vanilla self-supervised framework, which optimizes the similarities between two views through just one image-level representation per view. This is compared to pre-training with our proposed *DiRL* and *Cellback* frameworks. The encoders in our frameworks are implemented with both ViT-Tiny (ViT-T, $d = 192$) and ViT-Small (ViT-S, $d = 384$) consisting of 5M and 22M parameters, respectively.

**Dataset and tasks:** For *slide-level classification*, we use the following datasets: 1) TCGA-Lung (Albertina et al., 2016; Kirk et al., 2016) at $5\times$, 2) TCGA-BRCA (Lingle et al., 2016) at $5\times$, 3) TCGA-PAAD (Raphael et al., 2017) at $5\times$, and 4) BRIGHT (Brancati et al., 2021a) at $10\times$. Note that our proposed pre-training is performed separately for each dataset followed by evaluating them for slide-level classification. This classification task comprises 1) TCGA-Lung: Lung Adenocarcinoma (LUAD) versus Lung Squamous Carcinoma (LUSC), 2) TCGA-BRCA: Invasive Ductal (IDC) verses Invasive Lobular Breast Carcinoma (ILC), 3) TCGA-PAAD: Basal versus Classical Pancreatic Ductal Adenocarcinoma, 4) two sub-tasks in BRIGHT: 3-class WSI-classification (non-cancerous, precancerous, and cancerous), and 6-class WSI-classification, termed as BRIGHT (3) and BRIGHT (6), respectively.

For *patch-level classification*, evaluations are performed on Chaoyang (Zhu et al., 2021) and MHIST (Wei et al., 2021) datasets, which contain localized annotation at crop-level. MHIST consists of two classes of colon cancer, whereas Chaoyang contains four classes of colon cancer.

Note that, for generating cell prior $P_c$, we employed HoVer-Net for TCGA-Lung (Graham et al., 2019) and due to computational limitations we employed Cellpose (Stringer et al., 2021) for the other three WSI datasets. Further details on implementation and dataset splits are provided in A.1.

### 4.1 SLIDE-LEVEL AND PATCH-LEVEL CLASSIFICATION

**Slide-level classification:** In Table 1, we show the slide-level classification results on the four datasets with tiny and small ViT models pre-trained using *Baseline*, *Cellback* (*DiRL* without disentangle block), and *DiRL* frameworks. It may be observed that for the Lung, BRCA, and PAAD datasets, *DiRL* consistently surpasses the *Baseline* (up to 6.9% relative accuracy gain) and *Cellback* (up to 3.3% relative accuracy gain) models for both ViT-T and ViT-S architectures. In all cases, both *DiRL* and *Cellback* considerably outperform the vanilla-DINO *Baseline* (accuracy and AUC). Interestingly, *DiRL-T* performs even better than *Baseline-S* in BRCA and PAAD, substantiating the importance of efficiently encoding diversified information even in smaller feature embedding ($d = 192$) in *DiRL*-T against inefficiently and sparsely encoding into a larger feature embedding ($d = 384$) in ViT-S. This paves the direction for efficiently encoding domain-information in smaller models. In the BRIGHT dataset, it is observed that *Cellback-S* achieves the best performance for both the sub-tasks. We anticipate this might be due to the higher magnification ($10\times$) of this dataset as compared to the others ($5\times$). Consequently there is much less contextual information in WSI-crops for BRIGHT dataset, resulting in most regions being either cell-dominant or non-cell dominant (see Fig. 14 in the Appendix). This potentially makes encoding spatial interplay especially noisy due to lower co-occurence of the different concepts in a WSI-crop, thus hurting pre-training with disentangle block in *DiRL* (discussed in more detail in Appendix A.5). Additional comparisons with SOTA SSL methods are provided in the Appendix A.2.

**Patch-level classification:** For evaluating the generalizability of the learned representations, we use BRCA pre-trained models and fine-tune them on MHIST (Wei et al., 2021) and Chaoyang (Zhu et al., 2021) datasets (because of visual similarities between breast and colon cancers (Bremond et al., 1984)). In Table 2, we report the 5-fold cross valida-

Table 1: Results for slide-level classification tasks. T denotes ViT-Tiny, and S denotes ViT-Small.

| Dataset
Metric | Lung
Acc, AUC | BRCA
Acc, AUC | PAAD
Acc, AUC | BRIGHT (3)
Acc, AUC | BRIGHT (6)
Acc, AUC |
|---|---|---|---|---|---|
| Baseline-T | 0.895, 0.959 | 0.896, 0.944 | 0.768, 0.661 | 0.625, 0.841 | **0.512**, 0.764 |
| Cellback-T | 0.906, 0.964 | 0.896, 0.938 | 0.803, **0.713** | **0.650**, 0.848 | **0.512**, 0.793 |
| DiRL-T | **0.906**, **0.965** | **0.927**, **0.957** | **0.821**, 0.708 | **0.650**, **0.859** | 0.500, **0.812** |
| Baseline-S | 0.911, 0.967 | 0.896, 0.947 | 0.821, 0.704 | 0.637, 0.835 | 0.475, 0.787 |
| Cellback-S | 0.921, 0.966 | **0.927**, **0.955** | 0.821, 0.719 | **0.675**, **0.854** | **0.525**, **0.813** |
| DiRL-S | **0.927**, **0.973** | **0.927**, 0.950 | **0.839**, **0.733** | 0.650, 0.852 | 0.500, 0.799 |

tion accuracy and AUC on the official test set. We observed that both our models (*Cellback* and *DiRL*) outperform the *Baseline* on the two datasets using ViT-T and ViT-S backbones. Relative to the *Baseline*, *DiRL* improves the accuracy by $1.4 - 2\%$ on MHIST and $0.8 - 1.1\%$ on the Chaoyang dataset. Similarly, it also improves the corresponding AUCs by $0.4 - 0.7\%$ and $0.3 - 0.4\%$, respectively.

**Comparison of *DiRL* with other dense pre-training methods:**
Note that our pre-training aligns with dense pre-training literature as we perform dense matching across two views through region-level and disentangled representations instead of just matching through one image-level representation. For fair comparison, we re-implement dense pre-training methods closely related to our research: 1) SelfPatch (Yun et al., 2022) and 2) EsViT (Li et al., 2021b). In addition to image-level matching as in DINO, Self-Patch enforces invariance against each patch/token and its neighbors, whereas EsViT enforces matching between all the corresponding patch-based tokens across views. Note that we use ViT-S as the encoder for both the techniques. In Table 3, we showcase the results for SelfPatch and EsViT for slide-level classification tasks on all four datasets.

Table 2: Results for crop-level classification tasks.

| Dataset | MHIST | Chaoyang |
|---|---|---|
| Metric | Acc, AUC | Acc, AUC |
| Baseline-T | 0.867, 0.935 | 0.819, 0.942 |
| Cellback-T | 0.868, 0.934 | 0.823, 0.942 |
| DiRL-T | **0.884**, **0.942** | **0.828**, **0.945** |
| Baseline-S | 0.885, 0.945 | 0.830, 0.946 |
| Cellback-S | 0.896, **0.952** | 0.831, **0.950** |
| DiRL-S | **0.897**, 0.949 | **0.837**, **0.950** |

We find that our *DiRL*-based models perform on par with Es-ViT on BRIGHT and BRCA dataset, and significantly outperform on the Lung and PAAD datasets. Whereas SelfPatch performs significantly worse in most tasks, possibly because neighborhood token invariance hardly exists in pathology images unlike for well-defined objects in natural images. Thus, our domain-inspired dense matching in *DiRL* shows consistent improvements for slide-level classification compared to the other densely pre-trained models.

Table 3: Comparison of *DiRL* with existing dense pre-training SSL methods (SelfPatch and EsViT).

| Dataset | Lung | BRCA | PAAD | BRIGHT (3) | BRIGHT (6) |
|---|---|---|---|---|---|
| Metric | Acc, AUC | Acc, AUC | Acc, AUC | Acc, AUC | Acc, AUC |
| Baseline-S | 0.911, 0.967 | 0.896, 0.947 | 0.821, 0.704 | 0.637, 0.835 | 0.475, 0.787 |
| SelfPatch-S | 0.724, 0.784 | 0.886, 0.926 | 0.821, 0.728 | 0.512, 0.683 | 0.362, 0.636 |
| EsViT-S | 0.916, 0.967 | **0.927**, 0.954 | 0.768, 0.431 | **0.675**, **0.857** | **0.525**, 0.790 |
| Cellback-S | 0.921, 0.966 | **0.927**, **0.955** | 0.821, 0.719 | **0.675**, 0.854 | **0.525**, **0.813** |
| DiRL-S | **0.927**, **0.973** | **0.927**, 0.950 | **0.839**, **0.733** | 0.650, 0.852 | 0.500, 0.799 |

## 4.2 Analysis of learned Attention

Here we demonstrate the de-sparsification of the learned attention of our *DiRL* pre-trained models. Recall that the aggregated attention associated with a token is represented by the sum of all values across its corresponding column in the $n \times n$ self-attention matrix. The sum of the aggregated attention values of all tokens ($n$) should be $n$. Due to this constraint, if the model attends to some tokens with high attention values, then the attention value associated with other tokens are reduced significantly. In Fig. 4, we plot the distribution of aggregated attention values of tokens from the last layer of the transformer encoder pre-trained by *Baseline* (vanilla DINO), *Cellback*, *DiRL*, EsViT, and SelfPatch. We then split the aggregated attention values in three bins: 0-0.5, 0.5-2, >2. The 0-0.5 and >2 bins indicate sparse attention learned

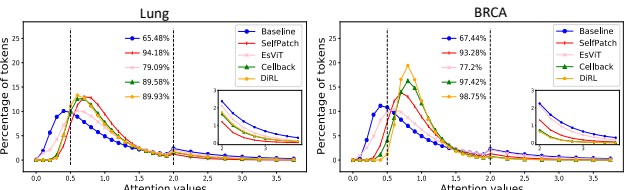

Figure 4: **Attention distribution plot.** The second bin (0.5-2) is the desired one. Here, the percentage values show the fraction of tokens with attention values in the desired range. The baseline method has a higher fraction of tokens in lower range and higher range sparse bins, which is not ideal for digital pathology applications.

through low and high concentrated attention values, respectively. Whereas the 0.5-2 range is the desired bin with moderate attention values that would lead to a de-sparsified attention map (and hence, optimal encoding of context-rich information). Plots of attention distribution of the tokens are illustrated individually on the test sets for Lung and BRCA. Plots for PAAD and BRIGHT are illustrated in Appendix A.6.

The plots show that *Baseline* trained with vanilla DINO has around 20-30% tokens in the lower range sparse bin (0-0.5) and around 8-10% in higher range sparse bin (>2). Whereas for our *Cellback* and *DiRL* models, fewer than 5% tokens lie in the lower range sparse bin, while 3-5% lie in higher range sparse bin. Importantly, our models yield significantly more diversified attention with more than 90% tokens in the desired bin (0.5-2) compared to that of 65% tokens for the *Baseline*. Interestingly, SelfPatch is able to diversify the transformer attention well, avoiding the sparse bins. However it still performs 2-20% lower than our models on various slide-level classification tasks . This might be due to the neighbor invariant self-supervision (refer to Yun et al. (2022)) being noisy in histopathology domain (as discussed in 4.1). EsViT consistently contains 10-15% more tokens compared to the *Baseline* in the desired bin. However it still contains much more tokens in the sparse bins compared to *DiRL* and *Cellback*. These observations justify our premise that dense matching could diversify the attention, which is crucial for learning representations for histopathology.

In Fig. 5, we visualize the attention overlay from models pre-trained using *Baseline* vanilla DINO, and our proposed *DiRL* and *Cellback*. The regions containing tumor cells are outlined in white, while those with necrosis and immune cells are outlined in yellow and green, respectively. It is evident that the *Baseline* model is sparsely attending the WSI crop, often ignoring crucial tumor cell-dominant regions. In contrast, our models are able to globally diversify attention. Bar plots show that almost all tokens have moderate attention values ranging from 0.5 to 2 in *DiRL*. In contrast, *Baseline* has a large number of tokens having very low attention (<0.5). Note that all the attention values > 1 are clipped to 1 for visualization. Additional visualizations are provided in A.6.

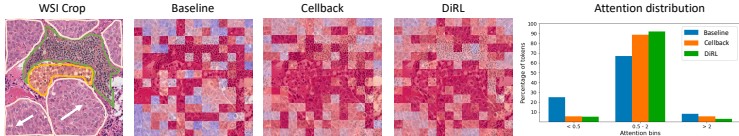

Figure 5: **Attention visualization.** Depicts the sparse attention by *Baseline*, and its subsequent de-sparsification by our methods on a representative lung cancer patch (detailed in A.6). Bar plot shows the percentage of tokens in the three bins. *Baseline* contains greater than 30% of tokens' attention values in the sparse bins; comparatively our method contains fewer than 10%.

### 4.3    ABLATION STUDIES

Here we study the utility of various components proposed in our framework. We perform our ablations on the Lung cancer dataset.

**Data efficiency:** We investigate the effect of pre-training with different fractions (20-100%) of total training data. As seen in Fig. 6, the gain in both AUC and accuracy is around 6% when *DiRL*-based models are pre-trained with significantly less data

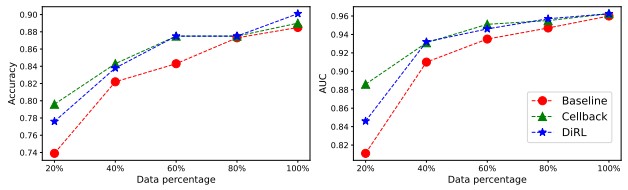

Figure 6: **Data efficiency study.** Illustrates the effect of pre-training with different amount of training data.

(20% data). These empirical findings show the importance of *DiRL* especially in low data regimes, for e.g. rare cancers. Other ablations including: 1) *Baseline* with an additional layer, 2) Effect of cell segmentation pipeline, and 3) Adaptation of *DiRL* with SimCLR SSL, are discussed in A.3.

## 5    CONCLUSION

In this work, we present a crucial requirement of tailoring SSL techniques for digital pathology through our insightful observation about the sparsity of attention. We proposed *DiRL*, a framework that densely encodes pathology characteristics and uses them in a dense matching objective for prior-guided pre-training. Through qualitative analysis, we showed that *DiRL* de-sparsifies the attention map, thus boosting the capabilities to encode diverse information in complex histopathology imaging. This was corroborated by consistent performance improvement on multiple slide-level and patch-level classification tasks by *DiRL*. We believe our work opens exciting avenues toward utilizing domain-specific concepts and instilling this domain knowledge in neural networks.

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

## A APPENDIX

We provide additional details regarding the following in this section:

- Dataset and Implementation details A.1
- Additional experiments and results A.2
- Additional ablation studies A.3
- Effect of WSI magnification on *DiRL* A.5
- Token-level analysis A.4
- Additional attention distributions and visualization A.6
- Disentangled block from pathological point of view A.7

### A.1 DATASET AND IMPLEMENTATION DETAILS

#### A.1.1 DATASETS

**WSI Datasets:** 1) *TCGA-Lung*: This dataset consists of 940 diagnostic digital slides from two sub-types of Lung cancer - Lung adenocarcinoma (LUAD) and Lung Squamous cell carcinoma (LUSC). We split the data into 748 training (391 LUAD, 357 LUSC) and 192 (96 LUAD, 96 LUSC) testing samples randomly. The WSI-crops (670K train, 150K test) are extracted at $5\times$ magnification. 2) *TCGA-BRCA*: This dataset consists 1034 diagnostic digital slides of two subtypes of Breast cancer - Invasive ductal carcinoma (IDC) and Invasive lobular carcinoma (ILC). We split the data into 937 training (747 IDC, 190 ILC) and 97 (77 IDC, 20 ILC) testing samples randomly. The WSI-crops (790K train, 90K test) are extracted at $5\times$ magnification. 3) *TCGA-PAAD*: This dataset consists of 168 diagnostic digital slides of two subtypes of Pancreatic cancer - Classical and Basal. We split the data into 112 training (93 Classical, 19 Basal) and 56 (43 Classical, 13 Basal) testing samples randomly. The WSI-crops (113K train, 60K test) are extracted at $5\times$ magnification. 4) *BRIGHT*: Comprises 703 (423 training, 80 validation, 200 testing) diagnostic digital slides. This dataset contains two sub-tasks: 3-class WSI classification and 6-class WSI classification tasks. For the first sub-task, the 3 classes are as follows - Non cancerous (PB+UDH), Pre-cancerous or Atypi-cal (ADH+FEA), and Cancerous (DCIS+IC). For the second sub-task, the 6 classes are as follows - Pathological Benign (PB), Usual Ductal Hyperplasia (UDH), Flat Epithelia Atypia (FEA), Atypical Ductal Hyperplasia (ADH), Ductal Carcinoma in Situ (DCIS), and Invasive Carcinoma (IC). The BRIGHT challenge contains train, validation, and test splits. Since the challenge is not active now, labels for the test set are not available. Therefore, we reported our results for this dataset on its validation set as our test set. Class-wise data split can be found here [1]. The WSI-crops (1.24M train, 0.2M test) are extracted at $10\times$ magnification.

**Patch Datasets:** 1) *MHIST* (Wei et al., 2021): Consists of 3152 images of colon with tasks to classify the type of colorectal polyps into two types, benign and pre-cancerous. All the image resolutions are of $224 \times 224$ pixels. 2) *Chaoyang* (Zhu et al., 2021): Consists of 6160 patches of size $512 \times 512$ pixels from Colon cancer divided into four classes - normal, serrated, adenocarcinoma, and adenoma. These patches are resized to $224 \times 224$ pixels in our experiments.

For these two patch datasets, we split their official training sets into a 5 fold cross validation sets. We train on 4 folds, validate on 1 fold and test on their official test sets. Thus, we report our results (accuracy and AUC) as a mean of 5 fold cross validation trials.

#### A.1.2 IMPLEMENTATION DETAILS.

For all our experiments, $224 \times 224$ sized crops are extracted from WSIs. We set the patch size for vision transformer input to $p = 16$. Therefore, the number of tokens per WSI crop are $n = 196$. For ViT-Tiny (ViT-T), the embedding dimension $d = 192$, whereas for ViT-small (ViT-S), $d = 384$.

*Pre-training:* For pre-training with DINO, we follow the hyper-parameter initialization from their source code (Caron et al., 2021). We use a batch size of 256. In pre-training *DiRL*, we set the loss weighting factors $\lambda_1 = 0.5$ and $\lambda_2 = \frac{0.1}{4}$, whereas for *DiRL* without disentangle block (*Cellback*),

---

[1]https://research.ibm.com/haifa/Workshops/BRIGHT/

we set $\lambda_1 = 0.5$. We use two different projection heads of default output sizes (65536) (Caron et al., 2021) for region representations $f_c$ and $f_b$, and four different projection heads of smaller output size (4096) for disentangled spatial interplay features. For pre-training with i) SimCLR, we adopted the implementation from Li et al. (2021a) with a batch size 512, ii) EsViT, implementation is adopted from Li et al. (2021b), and iii) SelfPatch, implementation is adopted from Yun et al. (2022). Note that pre-training is only performed on training samples of WSI datasets. For slide-level classification tasks, all models are pre-trained for 100 epochs on Lung and PAAD datasets, for 50 epochs on BRIGHT, and 30 epochs on BRCA. To study data-efficiency plot in Fig. 6, all the models are pre-trained for 50 epochs on the Lung dataset.

*Multiple instance learning:* We use DSMIL (Li et al., 2021a) for slide-level classification throughout this study. For training DSMIL, we use a learning rate of $2e^{-4}$, and weight decay of $5e^{-2}$. Batch size is set to 1 to handle variable bag size for each WSI $\mathcal{W}$. Other hyper-parameters such as number of epochs, and train-validation split ratio are kept consistent with Li et al. (2021a). Note that training samples from the WSI datasets are split into train and validation for MIL training. We hope to explore the impact of other MIL frameworks such as Ilse et al. (2018); Shao et al. (2021); Lu et al. (2021b); Lerousseau et al. (2021); Chen et al. (2022); Zhang et al. (2022); Pinckaers et al. (2020) on our *DiRL* learned features in future.

*Patch classification:* In our experiments of fine-tuning for patch classification, an average pooling layer (for averaging the tokens) followed by a fully connected layer is placed on top of the pre-trained transformer-encoder backbone. For all experiments on MHIST dataset, we use a learning rate of $3e^{-4}$, weight decay of $1e^{-2}$, and batch size of 128. We train the network for 40 epochs and decay the learning rate by 0.1 at epoch 20 and epoch 30. For all experiments on the Chaoyang dataset, we use a learning rate of $1e^{-4}$, weight decay of $1e^{-2}$, and batch size of 128. We train the network for 45 epochs and decay the learning rate by 0.1 at epoch 20, 30 and 40.

## A.2 ADDITIONAL EXPERIMENTS AND RESULTS

**Comparison with HIPT.** HIPT (Chen et al., 2022) is a recently proposed SSL framework tailored for digital pathology. We compare the performance of our *DiRL*-based models with HIPT on the datasets common to both studies, i.e, Lung and BRCA. HIPT consists of eight transformer blocks of ViT-S followed by four blocks of ViT-tiny. For the model size to be comparable to HIPT parameter size, we adopted a similar model architecture as HIPT, termed with notation -H. Note that the two set of blocks in HIPT are trained in a hierarchical way, whereas we simply train all the 12 blocks together in an end-to-end manner. We remain consistent with their data split for prior-guided pre-training and MIL training. On Lung datasets, *Cellback* shows relative improvement of 3% in both accuracy and AUC compared to HIPT. Whereas on BRCA, *Cellback* significantly outperforms HIPT with relative improvement of more than 9% in accuracy and 6% in AUC. Note that HIPT is pre-trained on whole The Genome Cancer Atlas (TCGA), consisting of 10678 WSIs from 33 cancer types. In contrast, we just pre-trained for each dataset separately, thus using 10 times less WSIs.

Table 4: Comparison with HIPT (Chen et al., 2022)

| Dataset | Lung | | BRCA | |
| Metric | Acc | AUC | Acc | AUC |
|---|---|---|---|---|
| HIPT | 0.862 | 0.942 | 0.839 | 0.901 |
| Cellback-H | **0.888** | **0.971** | **0.917** | **0.958** |

**Comparison with other methods in BRIGHT challenge:** Here we compare F1-scores of our models with a few papers published as part of the BRIGHT challenge (Wentai et al. (2022); Zhan et al. (2022); Marini et al. (2022)), termed as Method - 1, Method - 2, and Method - 3 respectively, and the BRIGHT baseline presented by the challenge organizers in Brancati et al. (2021b). Note that BRIGHT challenge consists of WSI-level labels, as well as 3000+ annotated ROIs with an average size $2000 \times 2000$ pixels. Using them for supervision can naturally boost the performance. BRIGHT baseline doesn't use annotated ROIs. Method - 2 and Method - 3 utilized the ROIs for training their feature encoder. In contrast, Method - 1 presents two experiments, one with SSL on WSIs for feature encoder, and one with fully supervised training (FS) for training feature encoder. Following

this, for MIL, they experimented with both CLAM and Reformer for slide-level classification. For MIL framework in our experiments we adopted DSMIL througout this study for slide-level tasks, and as DSMIL is more similar to CLAM than a transformer-based Reformer, for fair comparison we included their experiments with CLAM. It can be observed that for methods not supervised with localized annotation at ROI-level, *Cellback* performs best for both 3-class and 6-class classification tasks in terms of F1-score. Compared to supervised counterpart, our *Cellback* still performs best for 6-class task, whereas just slightly lower than Method - 2 and Method - 3.

Table 5: F1-score on BRIGHT validation set.

| Dataset
Metric | BRIGHT (3)
F1-score | BRIGHT (6)
F1-score | ROIs |
|---|---|---|---|
| BRIGHT baseline (Brancati et al., 2021b) | 0.580 | 0.390 | ✗ |
| Method - 1 (SSL + CLAM) (Wentai et al., 2022) | 0.642 | 0.412 | ✗ |
| Baseline-S | 0.632 | 0.467 | ✗ |
| Cellback-S | **0.663** | **0.494** | ✗ |
| DiRL-S | 0.625 | 0.467 | ✗ |
| Method - 1 (FS + CLAM) (Wentai et al., 2022) | **0.691** | **0.453** | ✓ |
| Method - 2 (Zhan et al., 2022) | 0.680 | 0.440 | ✓ |
| Method - 3 (Marini et al., 2022) | 0.650 | 0.450 | ✓ |

### A.3 ADDITIONAL ABLATION STUDIES

**Baseline with an additional layer.** For a fair comparison with *DiRL* pre-training which contains additional disentangle transformer block, we implement the *Baseline* ViT-S with an additional layer, i.e. a model with 13 transformer blocks. We explored two versions of this model: 1) only one DINO projection head from the $13^{th}$ layer (*Baseline*-S -13), 2) two DINO projection heads, one from the $12^{th}$ layer and another from the $13^{th}$ layer (*Baseline*-S -13$^*$). In Table 6, both these models perform sub-par to our proposed pre-training, confirming the importance of our domain-aware design choice in *Cellback* and *DiRL* over just adding more model parameters.

Table 6: Effect of additional layer in *Baseline*

| Dataset
Metric | Lung
Acc | 
AUC | BRIGHT (3)
Acc | 
AUC | BRIGHT (6)
Acc | 
AUC |
|---|---|---|---|---|---|---|
| Baseline-S | 0.911 | 0.967 | 0.637 | 0.835 | 0.475 | 0.787 |
| Baseline-S - 13 | 0.916 | 0.963 | 0.625 | 0.808 | 0.462 | 0.766 |
| Baseline-S - 13* | 0.916 | 0.966 | 0.637 | 0.850 | 0.462 | 0.762 |
| Cellback-S | 0.921 | 0.966 | **0.675** | **0.854** | **0.525** | **0.813** |
| DiRL-S | **0.927** | **0.973** | 0.650 | 0.852 | 0.500 | 0.799 |

**Effect of cell segmentation pipelines:** We evaluate *DiRL* framework with cell prior $P_c$ generated from three different cell segmentation pipelines namely HistomicsTK, Cellpose, and HoVer-Net on Lung cancer subtyping task in Table 7. HistomicTK is a python API which provides a handcrafted approach for cell segmentation based on Wu et al. (2004); Lowe (2004); Al-Kofahi et al. (2009). The other two pipelines constitute powerful deep learning based models. Among the three, HoVer-Net > Cellpose > HistomicTK in terms of segmentation performance (see Fig. 15, 16). It can be observed that models trained with the near-perfect segmentation pipeline (HoVer-Net) performs the best in slide-level classification compared to HistomicTK and Cellpose. It is noteworthy that for other two segmentation pipelines, our method still performs on par with the vanilla DINO *Baseline*, except for *Cellback* model using HistomicTK for cell prior. This exception is attributed to the poor segmentation quality by HistomicTK, thus infusing noise in matching regions between the views in pre-training. We believe that using HoVer-Net on other datasets will further boost their performance compared to using Cellpose in Table 1. In Fig. 7, we show the effect of cell segmentation pipelines on the de-sparsification of attention. Consistently across all the three segmentation pipelines, *Cellback* and *DiRL* achieve better de-sparsification compared to the *Baseline*.

Table 7: Effect of cell segmentation pipelines

| Dataset | | | **Lung** | | | |
|---|---|---|---|---|---|---|
| Segmentation | HistomicsTK | | HoVer-Net | | Cellpose | |
| Metric | Acc | AUC | Acc | AUC | Acc | AUC |
| Cellback-T | 0.890 | 0.960 | **0.906** | 0.964 | 0.901 | 0.960 |
| DiRL-T | **0.906** | 0.964 | **0.906** | **0.965** | 0.901 | 0.957 |
| Cellback-S | 0.901 | 0.971 | 0.921 | 0.966 | 0.911 | 0.969 |
| DiRL-S | 0.916 | 0.967 | **0.927** | **0.973** | 0.911 | 0.964 |

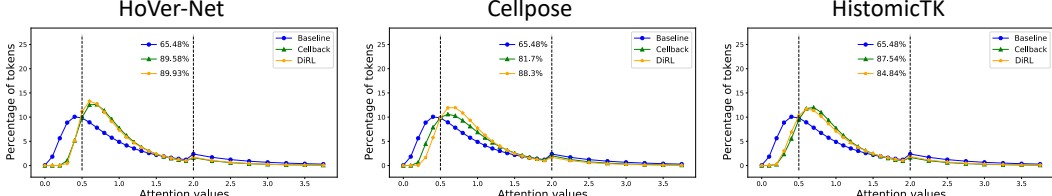

Figure 7: Attention distribution plot. Vertical lines separate the three defined bins. The percentage values show the percentage of tokens with attention values in the desired bin (0.5-2) for *Baseline*, *Cellback*, and *DiRL*.

**Adaptation of *DiRL* with other SSL frameworks:** In this study so far we adopted DINO framework for the self-supervision of the WSIs. However, *DiRL* framework can be incorporated with any self-supervised learning strategies. In Table 8, we demonstrate the performance of our proposed *Cellback* and *DiRL* representations pre-trained with either BYOL (Grill et al., 2020) or SimCLR (Chen et al., 2020) pipeline on the Lung dataset. We compare this adapted framework with *Baseline* models pre-trained with vanilla BYOL and vanilla SimCLR. Irrespective of SSL framework, *Cellback* and *DiRL* consistently outperform the *Baseline* in both Accuracy and AUC.

Table 8: Pre-training *DiRL* with other SSL frameworks. All the results are reported for models pre-trained for 20 epochs.

| SSL framework | Dataset | **Lung** | |
|---|---|---|---|
| | Metric | Acc | AUC |
| BYOL | Baseline-S | 0.744 | 0.822 |
| | Cellback-S | 0.796 | 0.858 |
| | DiRL-S | **0.802** | **0.866** |
| SimCLR | Baseline-S | 0.791 | 0.869 |
| | Cellback-S | **0.807** | 0.893 |
| | DiRL-S | 0.802 | **0.894** |
| DINO | Baseline-S | 0.802 | 0.891 |
| | Cellback-S | 0.812 | **0.904** |
| | DiRL-S | **0.823** | **0.904** |

**Effect of MixUp in vanilla SSL:** Another alternative to diversifying model attention would be to use stronger augmentation techniques. To test this, we evaluate the effect of MixUp in pre-training a ViT with SSL. For applying MixUp in self-supervision, we adopted i-mix (Lee et al., 2020) in the DINO framework. In Table 9, it can be observed, that this MixUp strategy improves the performance of the vanilla SSL due to its regularizing effects. However, this improvement is not attributed to the de-sparsification of the transformer attention weights (see Fig. 8). The complementary nature of our proposed approach and stronger augmentation techniques will be explored in future work.

Table 9: Pre-training DINO with MixUp

| Dataset | Lung | |
| Metric | Acc | AUC |
| --- | --- | --- |
| Baseline-S | 0.911 | 0.967 |
| Baseline-S w/ MixUp | 0.916 | 0.972 |
| Cellback-S | 0.921 | 0.967 |
| DiRL-S | **0.927** | **0.973** |

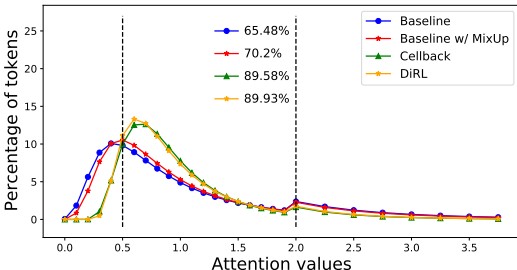

Figure 8: Attention distribution plot. Vertical lines separate the three defined bins. The percentage values show the percentage of tokens with attention values in the desired bin (0.5-2) for *Baseline*, *Baseline w/ MixUp*, *Cellback*, and *DiRL*.

**Multi-task adaptation of vanilla SSL:** An alternative approach of using cell segmentation as a prior could be to use the segmentation as an auxiliary task in self-supervised pre-training strategy. Consequently, in this section, we evaluate the impact of pre-training ViT with vanilla SSL jointly with a segmentation related auxiliary task. In order to avoid the use of a heavy decoder for segmentation, we instead design a new 'cell prediction task' to predict the number of cells present in each $p \times p$ patch of WSI-crop. A linear layer is applied on top of the ViT encoder for this task.

The joint optimization of vanilla SSL with the cell prediction task could hypothetically force the model to learn discriminative features from SSL and capture de-sparsification effects from the supervised loss ($L_{sup}$). We adopted the cell prediction supervised loss $L_{sup}$ with the *Baseline*. We find that the model with $L_{sup}$ could outperform the baseline model at early epochs (accuracy of 0.828 vs. 0.802 in baseline, AUC of 0.920 vs. 0.891 in baseline). However, with later training epochs, the supervised loss does not augment the vanilla SSL pre-training (accuracy of 0.911 vs. 0.911 in baseline, AUC of 0.967 vs. 0.967 in baseline). Thus, the multi-task adaptation leads to better convergence at lower epochs but under-performs our DiRL pre-training strategy when trained for a longer training schedule (100 epochs).

We analyzed the effect of $L_{sup}$ on the attention distribution in Fig. 9. It reveals that although the supervised loss helps to de-sparsify attention to an extent, the de-sparsification is still sub-par compared to *Cellback* and *DiRL*.

### A.4 TOKEN-LEVEL ANALYSIS

**Disentanglement of cellular and non-cellular regions.** To investigate whether *DiRL* implicitly distills the information regarding cell and back tokens into pre-trained model, we generate t-SNE plot from cell tokens and back tokens before their average pooling in the transformer encoder. For both token types, 30K tokens are randomly sampled from WSI-crops in the test set. We then fit separate 2-D Gaussians on the t-SNE points of cell tokens and back tokens (see Fig. 10), and measure the Jensen–Shannon divergence (JS) (Lin, 1991) and Bhattacharya distance (BD) (Bhattacharyya, 1943) between the two distributions. It can be observed in Table 10 that *DiRL > Cellback > Baseline*, in both JS and BD, which implies that *DiRL* can better separate the cell tokens from the back tokens, followed by *Cellback*, while the *Baseline* performs worst. This analysis provides us fur-

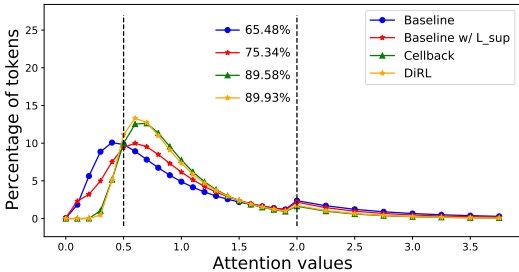

Figure 9: Attention distribution plot. Vertical lines separate the three defined bins. The percentage values show the percentage of tokens with attention values in the desired bin (0.5-2) for *Baseline*, *Baseline w/ $L_{sup}$*, *Cellback*, and *DiRL*.

ther insights that a model pre-trained with *DiRL* retains domain-specific information on cellular vs. non-cellular regions.



Figure 10: t-SNE plot of cell and back tokens.

Table 10: **Separability of cell-back tokens.** Jensen–Shannon divergence (JS) and Bhattacharya distance (BD) metrics are employed to measure the overlap of distribution of cell and back tokens mapped on t-SNE plot in Fig. 10. Higher values mean better separability.

|            | JS   | BD   |
|------------|------|------|
| Baseline-S | 1.04 | 0.26 |
| Cellback-S | 1.20 | 0.36 |
| DiRL-S     | **1.30** | **0.43** |

## A.5 EFFECT OF MAGNIFICATION

Fig. 14 illustrates WSI-crops extracted at $5\times$, and it's magnified view at $10\times$ magnification tiled into four crops with same image size. The crops extracted at $5\times$ contains four times more visual field compared to that of the crops extracted at $10\times$, though with less resolution. From Fig. 14, it is evident that the crops from $5\times$ contain visually diverse information, consisting of various structures such as glands and cell clusters. Whereas crops from $10\times$ are much less diverse, with crops dominantly consisting of either cellular or non-cellular regions. Consequently, co-occurence of both the regions is limited in same size WSI-crops at higher magnification like $10\times$ compared to the ones at $5\times$. Limited co-occurrence of both regions in a satisfactory quantity, could potentially lead to sub-optimal representation learning of spatial interplay patterns between different regions. We attribute *Cellback's* superior performance on BRIGHT dataset to this observation (more details rather than context). Hence, we conclude that utilizing the disentangle block is more favorable when operating at lower magnification like $5\times$. Thus *DiRL* is suitable when downstream tasks require lower magnification to focus at diverse spatial structures such as glands, tumor infiltrating lymphocytes (TILs), and their spatial interplay in the tumor microenvironment. This could be a possible reason

for *DiRL* being best for datasets processed at lower magnification (Lung, BRCA, PAAD, MHIST, Chaoyang) (see Table 1).

## A.6 ADDITIONAL ATTENTION DISTRIBUTIONS AND VISUALIZATIONS

In Fig. 11, we illustrates the distribution of learned attention as discussed in 4.2, for PAAD and BRIGHT datasets. *Baseline* model has around 35-40% tokens in lower range (0-0.5) and higher range (>2) sparse bins, while *DiRL* and *Cellback* contain less than 10% tokens in sparse bins. There observations on PAAD and BRIGHT are consistent with our observation of attention distribution on Lung and BRCA (refer to 4.2). Surprisingly, though the model pre-trained with EsViT on PAAD dataset contains all the tokens in the desired bin with uniform attention values close to one, it performs worse for slide-level classification task compared to *DiRL*, *Cellback*, and even *Baseline* (see Table 1). This shows that although maximum encapsulation of tokens in the desired bin is helpful, to achieve optimum performance they still need to be well distributed across the desired bin which is achieved by our *DiRL* models.

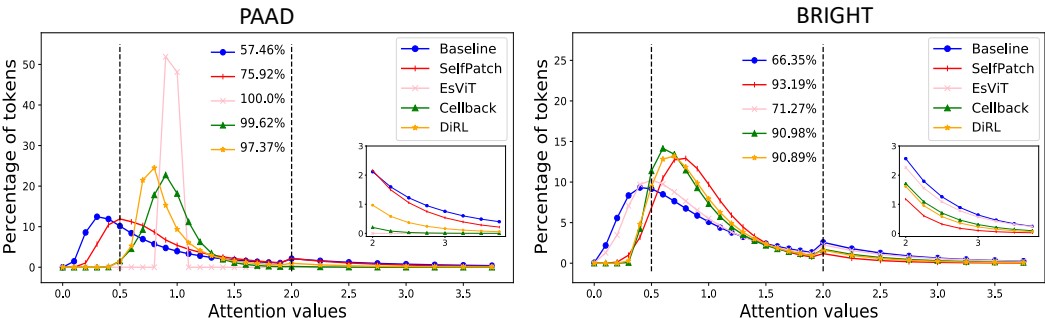

Figure 11: **Attention distribution plot on PAAD and BRIGHT dataset.** The second bin (0.5-2) is the desired one. Here, the percentage values show the fraction of tokens with attention values in the desired range. Similar to the plots in fig. 4, the *baseline* method has a higher fraction of tokens in lower range and higher range sparse bins, which is not ideal for digital pathology applications.

In Fig. 12, we visualize the attention overlay from models pre-trained using *Baseline* vanilla DINO, and our proposed *DiRL* and *Cellback*. We mark various regions such as region dominated by tumor cell, immune cells. It can be observed that for Lung cancer, model predominantly focus on immune cell region, while ignoring much of the tumor cells region. Oppositely in BRCA samples, model attends to tumor cells region mainly, while ignoring the immune cells. This could result in loss of crucial information in encoding the representation by *Baseline*. In contrast, *Cellback* and *DiRL* are able to adequately attend to each such region, thus encoding more informative representations.

So far, in qualitative attention analysis (see Fig. 5 and Fig. 12), we visualized the aggregated attention being given to the tokens at WSI-crop level. In Fig. 13, we now visualize the *interaction relationship* between a given query token with other tokens in the WSI-crop. We illustrated how a given query (marked with a yellow bounding box) interacts with (attends to) other tokens by visualizing the row of an attention matrix corresponding to the query. Our findings in query-level interaction analysis is consistent with that of previously shown WSI-crop level attention analysis, i.e, the query token sparsely attends to different tokens in baseline (vanilla SSL). In contrast, our pretraining alleviates this problem and densely interacts the query token with other tokens in the image.

## A.7 DISENTANGLED BLOCK FROM PATHOLOGICAL POINT OF VIEW

In this study, we aim to model the interaction between the cellular (comprising various types of cells) and non-cellular regions (comprising stroma, smooth muscle region, fat, etc). In pathology, the interaction between various entities (nuclei, stroma, glands, etc.) has been found to have clinical significance (Saltz et al., 2018; Diao et al., 2021; Zormpas-Petridis et al., 2021). For modeling the interaction between cellular and non-cellular regions, we believe that disentangling both the regions followed by explicitly encoding the inter-intra region interaction is necessary. Otherwise

without disentanglement, encoding the inter-intra region interaction would not be precisely achievable, pertaining to the fact that average pooling in vision transformers could potentially dilute these token-level crucial signals. Future directions could delve into utilizing more refined entities (such as immune cells region, tumor regions, glands, necrotic region, and stroma) and quantifying their mutual interactions, thus better guiding the neural network to learn intricacies of digital pathology.

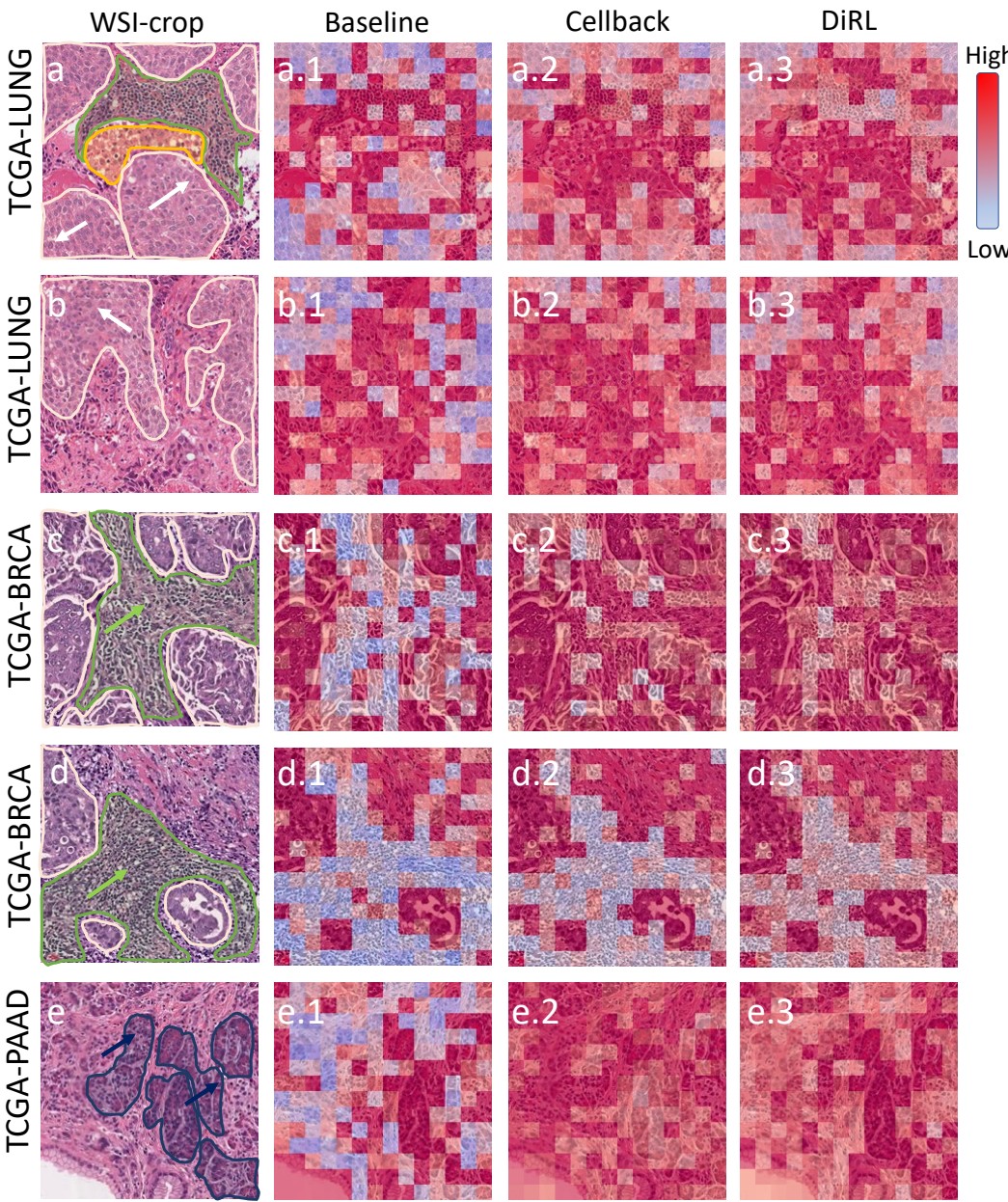

Figure 12: **Attention visualization.** a and b shows an example 5× patches (size 224 × 224) from lung cancer, c and d shows an example 5× patch (size 224 × 224) from breast carcinoma BRCA, e shows an example 5× patch (size 224 × 224) of a normal region from the pancreatic adenocarcinoma dataset. In column 2, (*.1) shows the *Baseline* model attention maps of the patches in column 1. In column 3, (*.2) shows the *Cellback* model attention maps of the patches in column 1. In column 4, (*.3) shows the *DiRL* attention maps of the patches in column 1. In a, b, c, and d, the tumor regions have been annotated in white, lymphocytic regions in green and necrosis in yellow. In the first and second rows, the arrows indicate the tumor regions which were sparsely attended to by the baseline as compared to our models. Though the baseline model attends well to the lymphocytic regions in a, it fails to densely attend to the important tumor areas. In the third and fourth row, the arrows indicate the lymphocytic regions which were sparsely attended to by the baseline as compared to our models. The tumor regions, however, exhibit high attention for all the models. In the fifth row, the baseline models sparsely attend to many of the acinar tissue regions (blue regions). In contrast, our model diversifies the attention over all the acinar regions.

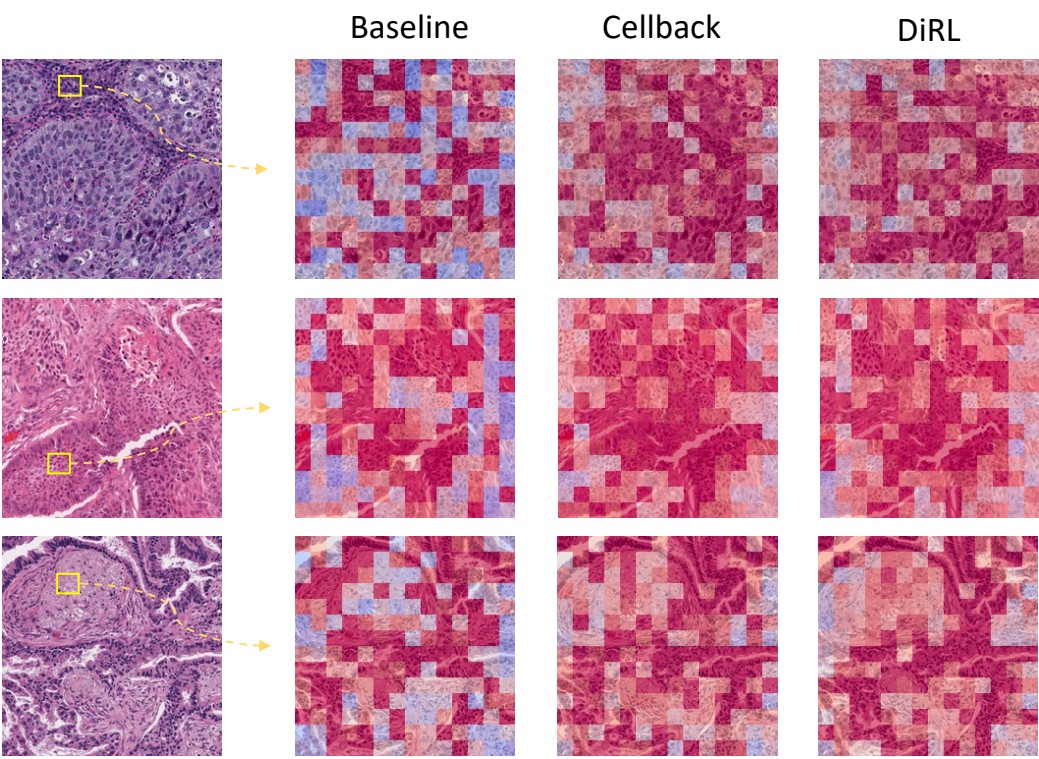

Figure 13: **Tokens Interaction Visualization.** Query token is marked in yellow bounding box. The interaction of the query token with all other tokens in the WSI-crop is illustrated for *Baseline*, *Cellback*, and *DiRL* . It can be observed that, in *Cellback* and *DiRL*, the query token interacts more densely to various tokens in the image compared to the *Baseline*.

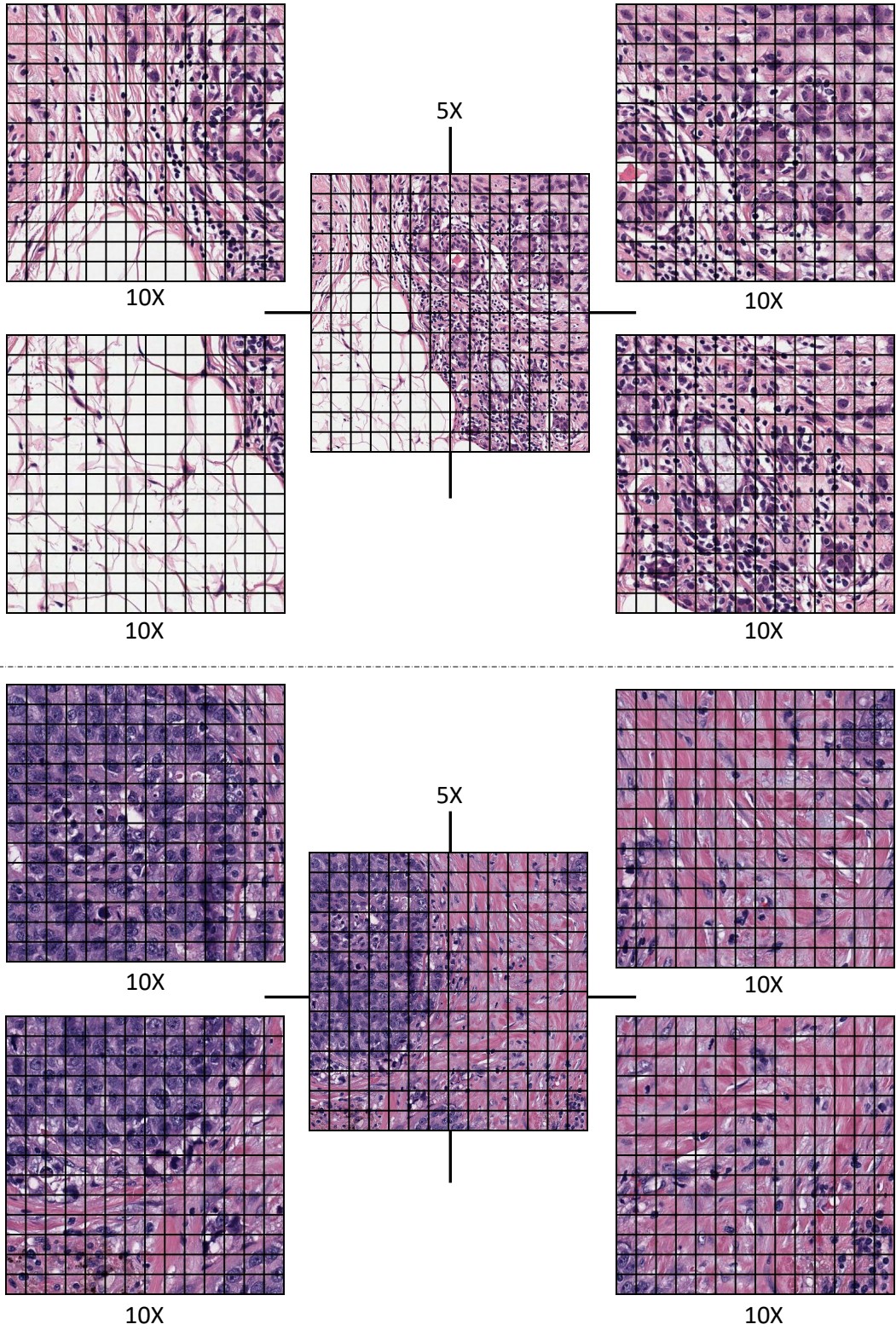

Figure 14: Illustration of WSI-crops extracted at 5× and their magnified view at 10× magnification tiled into four images with same size as of WSI-crops at 5×. The crops at 5× contain diverse phenotypic information such as cluster of cells, glands, and stroma. In contrast, the crops at 10× are often limited in phenotypic diversity of these structures.

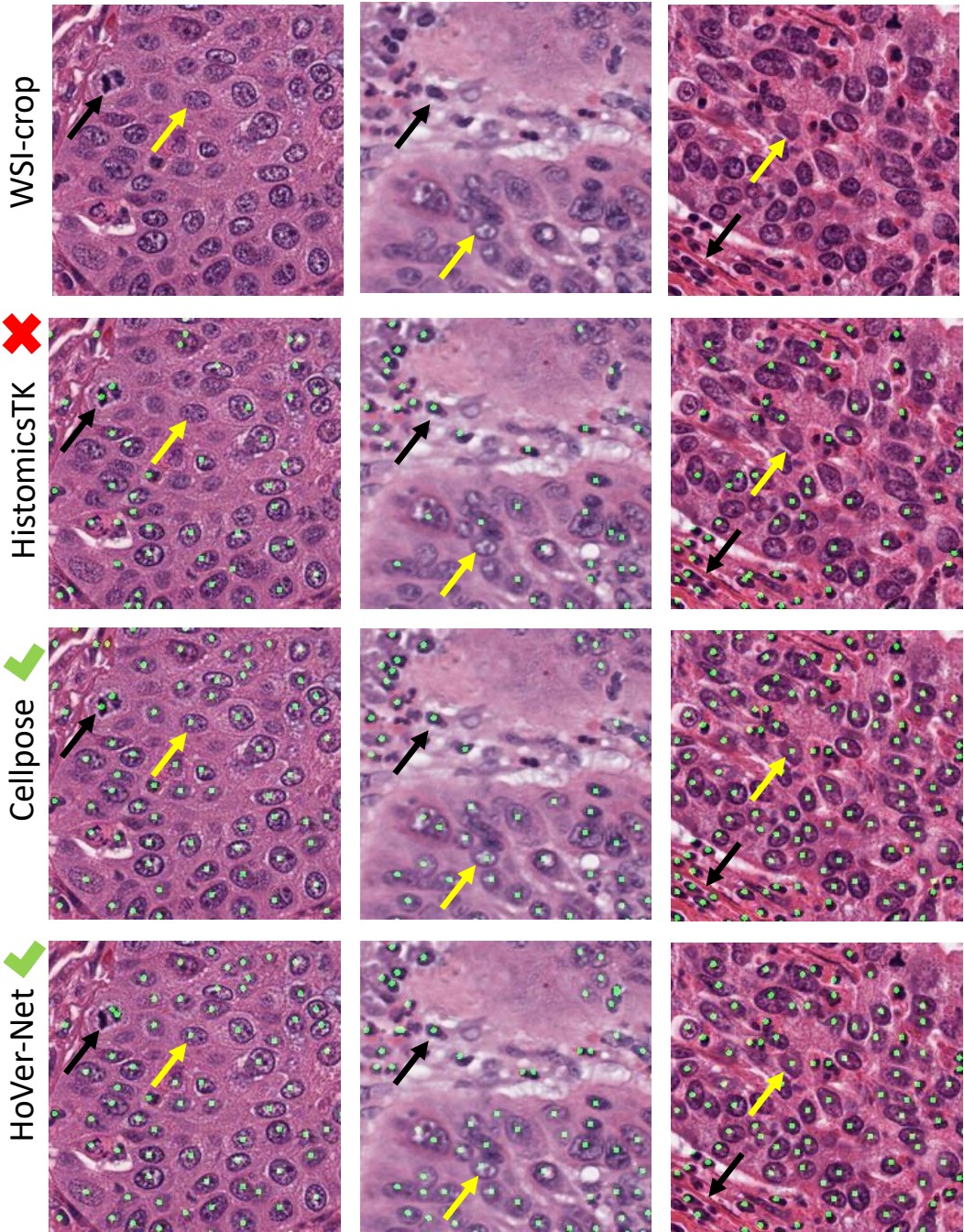

Figure 15: **Cell segmentation performance.** Illustrations of WSI-crops, and the output of cell segmentation from HistomicsTK, Cellpose, and HoVer-Net. Black arrows denote normal cells/lymphocytes, and yellow arrows denote large tumorous cells. HistomicsTK misses numerous tumorous cells, while adequately detecting the normal ones. The more powerful deep-learning pipelines, Cellpose and HoVer-Net, are able to capture both type of cells with greater precision.

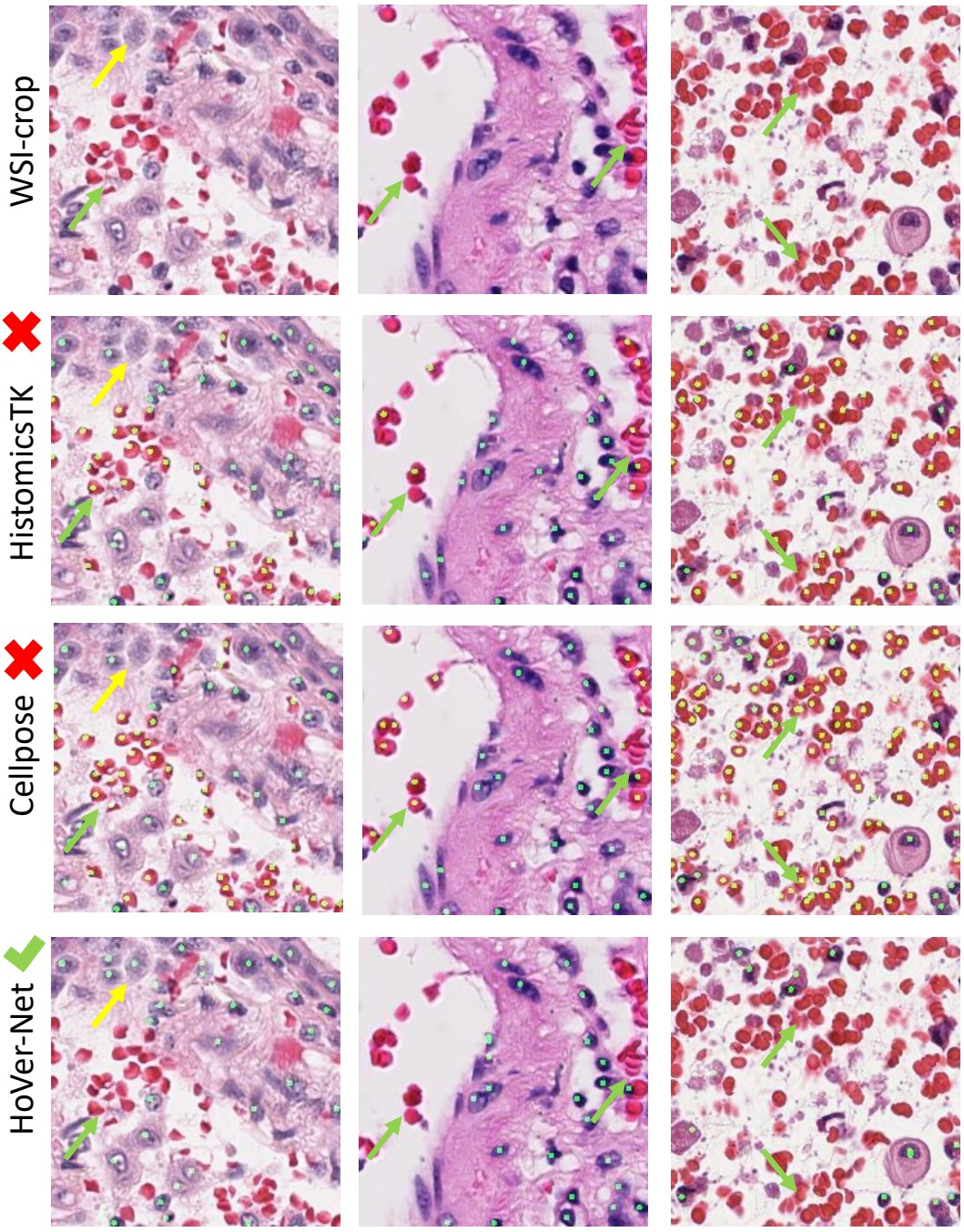

Figure 16: Illustrations of WSI-crops, and the output of cell segmentation from HistomicsTK, Cellpose, and HoVer-Net. Green arrow denotes blood cells, and yellow arrow denotes large tumorous cells. Note that the arrangement/distribution of blood cells is not implicated in tumor phenotyping. Both HistomicsTK and Cellpose segment the blood cells, while HoVer-Net avoids them to a large extent. This example shows that HoVer-Net is preferable.

