# OpenReview forum: "Attention De-sparsification Matters: Inducing Diversity in Digital Pathology Representation Learning"
_ICLR.cc/2023/Conference — Submitted to ICLR 2023_

### Official Review · Reviewer_6NiW · 2022-10-23

**Confidence:** 4
**Correctness:** 3
**Technical Novelty And Significance:** 3
**Empirical Novelty And Significance:** 2
**Recommendation:** 6

**Clarity, Quality, Novelty And Reproducibility:**

The overall quality of the manuscript is good. The idea of injecting prior-based attention in the self-attention module is novel, though the design could be improved and further investigated. Nevertheless, I believe it should be easy to reproduce the proposed method.

**Strength And Weaknesses:**

Strength:
+ The manuscript is overall well-written and easy to follow
+ The idea of injecting prior-based attention into the self-attention module is novel and seems effective, at least for pathological image-related applications.
+ Many public datasets are employed here for the experiments

Weakness:
- I am concerned about the comparison study, considering only baselines are compared. If the proposed attention add-on should be regarded as a general approach, other self-supervised learning frameworks/networks should be considered in the comparison study as well. Otherwise, SOTA results on these utilized datasets could also be reported to get a sense of how the proposed framework generally works.
- The definitions of M_self and M_cross are not well-introduced and discussed. I am unsure about the philosophy of setting the values to either 0 or minus infinite. Moreover, the M_self and M_cross are added to qk^T directly without considering the scale of these two vectors. Is it better to be a weighted sum?
- How the accuracy of M_self and M_cross will affect the overall training performance should also be investigated and discussed.

**Summary Of The Paper:**

The authors proposed manipulating the attention learned during self-supervised learning toward more meaningful regions (cells against the background). Precomputed masks of cells are injected into the self-attention modules, i.e., combined with the self-attention before the softmax. Attentions are further divided into groups of cells, backgrounds, and cross-cell-backgrounds. Several public datasets are employed here for the experiments and evaluation. Superior results of the proposed method are reported in comparison to self-supervised learning baselines (pre-trained with DINO).

**Summary Of The Review:**

The overall quality of the presented work is good, though there are some flaws in the introduction of the method and experimental settings. Adjusting the attention for medical images is essential in addition to the standard self-attention-based pre-training. The authors proposed a novel way toward that direction.

---

> ### Author Response · Authors · 2022-11-15
> **Response to Reviewer 3's critique**
>
> We thank Reviewer 3 for their constructive reviews and suggestions. Here, we specifically respond to Reviewer 3’s critiques. We have updated the paper and appendix, as appropriate.
>
> We would also like to point the Reviewer 3 to the author's response under :- “Terminology change: Di-SSL (Diversity inducing Self-Supervised Learning) changed to DiRL (Diversity inducing Representation Learning)”.
>
>
> Q1) I am concerned about the comparison study, considering only baselines are compared. If the proposed attention add-on should be regarded as a general approach, other self-supervised learning frameworks/networks should be considered in the comparison study as well. Otherwise, SOTA results on these utilized datasets could also be reported to get a sense of how the proposed framework generally works.
>
> - We thank the reviewer for this question regarding experimenting our proposed attention add-on with other SSL frameworks. We would like to point out that we had already evaluated our Cellback approach with SimCLR in appendix A.3 (Adaptation of DiRL with SimCLR). We showed that Cellback with SimCLR performs better than vanilla SimCLR. We have now considered the reviewer’s feedback and performed more exhaustive evaluations with other SSL frameworks;  specifically, we performed pre-training of ViT with vanilla SimCLR and vanilla BYOL. Similarly we adopted our Cellback and DiRL with both SimCLR and BYOL. We have added the results of pre-training with both the SSL frameworks and their comparison with DINO in Table 8 in Appendix A.3 (Adaptation of DiRL with other SSL frameworks). We observe that consistent with previous findings, both Cellback and DiRL outperform the vanilla SSL frameworks. These experiments solidify the importance of our contribution in diversifying the attention, irrespective of the self-supervised learning framework. Results are reported for models pre-trained for 20 epochs.
>
>
> Q2) The definitions of M_self and M_cross are not well-introduced and discussed. I am unsure about the philosophy of setting the values to either 0 or minus infinite. Moreover, the M_self and M_cross are added to qk^T directly without considering the scale of these two vectors. Is it better to be a weighted sum?
>
> - We apologize for the confusion caused. The philosophy of setting a few values to 0 and others to -\infty in the self-attention matrix to mask out interaction is motivated by the ‘Attention is All You Need’ paper [3]. This is a widely used strategy to implement self-attention for sequences with different lengths [5] and causal transformers [4]. In DiRL, we use this strategy to mask out certain undesired token interactions.  For M_self, inter-region (between cell and background) interactions are undesired, while for M_cross, intra-region interactions are undesired.
> Thus, this is implemented in the scaled dot-product attention by masking out (setting to -\infty) all values in the input of the softmax which correspond to undesired interactions.
>
> Q3) How the accuracy of M_self and M_cross will affect the overall training performance should also be investigated and discussed.
>
> - We would like to point out that M_self and M_cross are generated from the cell segmentation prior. Therefore, we refer the reviewer to appendix A.3 (Effect of cell segmentation pipelines). Here we have shown ablations on the effect of cell segmentation algorithms such as HistomicTK, Cellpose, and HoVer-Net on Cellback and DiRL.
>
>
> [1] Caron, Mathilde, et al. "Emerging properties in self-supervised vision transformers." Proceedings of the IEEE/CVF International Conference on Computer Vision. 2021.
>
> [2] Chen, Richard J., et al. "Scaling Vision Transformers to Gigapixel Images via Hierarchical Self-Supervised Learning." Proceedings of the IEEE/CVF Conference on Computer Vision and Pattern Recognition. 2022.
>
> [3] Vaswani, Ashish, et al. "Attention is all you need." Advances in neural information processing systems 30 (2017).
>
> [4] Chen, Lili, et al. "Decision transformer: Reinforcement learning via sequence modeling." Advances in neural information processing systems 34 (2021): 15084-15097.
>
> [5] Devlin, Jacob, et al. "Bert: Pre-training of deep bidirectional transformers for language understanding." arXiv preprint arXiv:1810.04805 (2018).

---

### Official Review · Reviewer_nwTd · 2022-10-24

**Confidence:** 4
**Correctness:** 3
**Technical Novelty And Significance:** 2
**Empirical Novelty And Significance:** 3
**Recommendation:** 6

**Clarity, Quality, Novelty And Reproducibility:**

Clarity: The paper is easy to follow and well organized.
Quality: Good.
Novelty: The observation is novel to me. The proposed solutions to de-sparsication is somewhat novel.
Reproducibility: The paper doesnot provide the codes with the submission.

**Details Of Ethics Concerns:**

No ethics concern.

**Strength And Weaknesses:**

Strength:
1. The observation is quite interesting and it makes sense.
2. The proposed strategies (domain-priors, pretext tasks..) is simple but make sense to de-sparsification the attention map.
3. The results on several datasets are promising. The authors indeed conduct many experiments and well describe the datasets and corresponding experimental results.

Weakness:
1. The paper only considers on SSL framework, what's the situation in other SSL frameworks?
2. It is hard to apply the proposed method on other tasks, because we first need a well trained segmentation model which are not easily achieved in many cases.
3. Suppose we have segmentation ground truth and train the segmentation model, mabye we can just design a multi-task (ssl+supervised segmentation) learning framework and then achieve the de-sparsification effects (segmentation force the network to learn details).
4. The claim "We believe our work opens exciting avenues toward utilizing biology-relevant concepts and enhancements in neural networks" is weird, where do you use biology-relevant concepts and enhancement?

Concerns:
1. I can imagine that the SSL models (e.g., DINO) on imagenet classification tasks produces sparse attention features (object-centric). I think that's because the task (classify those images mainly need to focus on the object) and data (object-centric). However, if the data is non-object-centric (for example, pathology slides), then the task (if well trained) should be able to force the model to learn more dense features, since it is these dense features which can contribute to correct classification. If it is in such a case, do we still need the de-sparsification step?
2. Maybe some augmentation which are well designed for non-object-centric data can better solve this problem. I believe the sparsification is due to the data and the task (classification), so if we use some kind of augmentation to generate the two views (considering the dense essence of the data), the SSL should be able to learn the de-sparsification features.

**Summary Of The Paper:**

This paper mentioned an observation: the vanilla SSL (for example, DINO) usually learned sparse features, which is suitable for object-centric tasks but not good for non object-centric tasks. As a result, the paper proposed using domain-priors together with several pretext tasks to learn de-sparsification features. The experiments on several datasets (slide-level/patch-level classification tasks) show the effectiveness of the proposed method.

**Summary Of The Review:**

This is an interesting paper to solve sparse feature problem in SSL for pathology images. Generally, I'm interested in the observation part and fine with the proposed method (I agree the proposed method can largely alleviate the problem). However, I donot think this paper touches the essence of the phenomenon, which I think is due to data and the task (and we can use certain data augmentation to generate better views and thus enforce the model to learn better "dense" features).

---

> ### Author Response · Authors · 2022-11-15
> **Response to Reviewer 2's critique - [part 1/3]**
>
> We thank Reviewer 2 for their constructive reviews and suggestions. Here, we specifically respond to Reviewer 2’s critiques. We have updated the paper and appendix, as appropriate.
>
> We would like to point R2 to the author's response under :- “Terminology change: Di-SSL (Diversity inducing Self-Supervised Learning) changed to DiRL (Diversity inducing Representation Learning)”.
>
>
> Q1) The paper only considers one SSL framework, what's the situation in other SSL frameworks?
>
> - We thank the reviewer for this excellent question regarding the generalizability of our method. We would like to point out that we had already evaluated our Cellback approach with SimCLR in appendix A.3 (Adaptation of DiRL with SimCLR). We showed that Cellback with SimCLR performs better than vanilla SimCLR. We have now considered the reviewer’s feedback and performed more exhaustive evaluations with other SSL frameworks;  specifically, we performed pre-training of ViT with vanilla SimCLR and vanilla BYOL. Similarly we adopted our Cellback and DiRL with both SimCLR and BYOL. We have added the results of pre-training with both the SSL frameworks and their comparison with DINO in Table 8 in Appendix A.3 (Adaptation of DiRL with other SSL frameworks). We observe that consistent with previous findings, both Cellback and DiRL outperform the vanilla SSL frameworks. These experiments solidify the importance of our contribution in diversifying the attention, irrespective of the self-supervised learning framework. Results are reported for models pre-trained for 20 epochs.
>
> Q2) It is hard to apply the proposed method on other tasks, because we first need a well trained segmentation model which is not easily achieved in many cases.
>
> - We agree with the reviewer that well trained segmentation models might be hard to achieve in niche tasks such as crowd counting [10], etc. However, significant progress in the last few years in the field of digital pathology has produced a plethora of cell segmentation tools such as HoVer-Net and Cellpose, which are very robust and generalizable [3,4,5,6]. HoVer-Net is supervised with 200,000 annotated nuclei from 19  organs, whereas Cellpose is supervised with a dataset of over 70,000 segmented cells. This large-scale training from various data sources makes these tools robust to directly employ on a diverse set of pathology slides. Previous studies [3,4,5,6] have also directly built on top of the cell segmentation output of these tools.
>
> - Also note that following pre-training, our method does not require the cell segmentation prior at inference. This is one of the strengths of our approach and makes it scalable for deployment.
>
> Q3) Suppose we have segmentation ground truth and train the segmentation model, maybe we can just design a multi-task (ssl+supervised segmentation) learning framework and then achieve the de-sparsification effects (segmentation force the network to learn details).
>
> - We thank the reviewer for this insightful idea. This is potentially an alternative approach to diversify the attention and force the model to learn details. To test this out, we have now pre-trained a ViT with vanilla SSL jointly with a segmentation related auxiliary task:
>
>    In order to avoid the use of a heavy decoder for segmentation, we instead design a new ‘cell prediction task’ to predict the number of cells present in each 16 X 16 patch of WSI-crops. A linear layer is applied on top of the ViT encoder for this task. The joint optimization of vanilla SSL with the cell prediction task could hypothetically force the model to learn discriminative features from SSL and capture de-sparsification effects from the supervised loss. We have added this experiment of utilizing this supervised loss to pretraining of vanilla DINO in the appendix A.3 (Multi-task adaptation of vanilla SSL). We find that the model with this supervised loss could outperform the baseline model at early epochs (accuracy of 0.828 vs. 0.802 in baseline). However, with later training epochs, the supervised loss does not augment the vanilla SSL pre-training (accuracy of 0.911 vs. 0.911 in baseline). Thus, the multi-task adaptation leads to better convergence at lower epochs but under-performs our DiRL pre-training strategy when trained for a longer training schedule (100 epochs). Our attention distribution analysis (see Figure 9 in the appendix) reveals that although the supervised loss helps to de-sparsify the attention to an extent, the de-sparsification is still sub-par compared to Cellback and DiRL.

---

> > ### Author Response · Authors · 2022-11-15
> > **Response to Reviewer 2's critique - [part 2/3]**
> >
> > Q4) The claim "We believe our work opens exciting avenues toward utilizing biology-relevant concepts and enhancements in neural networks" is weird, where do you use biology-relevant concepts and enhancement?
> >
> > - We regret the confusion. We have revised the manuscript with a more well structured claim - “We believe our work opens exciting avenues toward utilizing domain-specific concepts and instilling this domain knowledge in neural networks”.
> >
> > - We would like to clarify here that the domain-specific knowledge we use is the fact that for tumor subtyping (a particular class of biological problem), transformer attention cannot be sparse; it needs to be dense and diverse. To achieve attention diversity, we further use domain concepts such as cells. This could be readily extended to other pathology-specific structures such as stroma, epithelia, etc [8, 9].
> >
> > Q5) I can imagine that the SSL models (e.g., DINO) on imagenet classification tasks produce sparse attention features (object-centric). I think that's because of the task (classify those images mainly need to focus on the object) and data (object-centric). However, if the data is non-object-centric (for example, pathology slides), then the task (if well trained) should be able to force the model to learn more dense features, since it is these dense features which can contribute to correct classification. If it is in such a case, do we still need the de-sparsification step?
> >
> > - That is an excellent point; we had the same expectation as the reviewer. However when we applied vanilla SSL to pathology, we were proven wrong. Given that pathology slides are non-object-centric, the pretext task should be able to force the model to learn more dense features. However the limitation is in the pretraining strategy where ‘no spatial constraint exists over matching the distribution of representations across the views’. Consequently, the learned model localizes most of its attention to a small fraction of regions (see Figure 4 in the main paper), leading to suboptimal representation learning.
> >
> > - Our empirical attention analysis in Section 4.2 of the paper proves that vanilla SSL collapses to match certain prominent patterns of the dataset, while ignoring the other regions. This is a particularly undesirable property of vanilla SSL in the pathology domain, and therefore motivates our de-sparsification strategy.

---

> > > ### Author Response · Authors · 2022-11-15
> > > **Response to Reviewer 2's critique - [part 3/3]**
> > >
> > > Q6) Maybe some augmentation which are well designed for non-object-centric data can better solve this problem. I believe the sparsification is due to the data and the task (classification), so if we use some kind of augmentation to generate the two views (considering the dense essence of the data), the SSL should be able to learn the de-sparsification features.
> > >
> > > - To address the reviewer's concern we performed an experiment with stronger data augmentation in vanilla SSL pretraining. Following the i-mix paper [7], we adopted the MixUp strategy in the DINO framework to force the model to focus on the dense information available. The i-mix paper [7] showed that MixUp performs better than CutMix in SSL; thus we utilized MixUp. We have included the results in Table 9 in Appendix A.3 (Effect of MixUp in DINO). We observe that this mixup strategy improves the performance of the vanilla SSL due to its regularizing effects. However, this improvement is not attributed to the de-sparsification of the transformer attention weights (see Figure 8 in the Appendix A.3). The complementary nature of our proposed approach and stronger augmentation techniques will be explored in future work.
> > >
> > >
> > > [1] Caron, Mathilde, et al. "Emerging properties in self-supervised vision transformers." Proceedings of the IEEE/CVF International Conference on Computer Vision. 2021.
> > >
> > > [2] Chen, Richard J., et al. "Scaling Vision Transformers to Gigapixel Images via Hierarchical Self-Supervised Learning." Proceedings of the IEEE/CVF Conference on Computer Vision and Pattern Recognition. 2022.
> > >
> > > [3] Lu, Wenqi, et al. "SlideGraph+: Whole Slide Image Level Graphs to Predict HER2 Status in Breast Cancer." Medical Image Analysis (2022): 102486.
> > >
> > > [4] Pati, Pushpak, et al. "Hierarchical graph representations in digital pathology." Medical image analysis 75 (2022): 102264.
> > >
> > > [5] Feng, Chao, Chad Vanderbilt, and Thomas Fuchs. "Nuc2vec: Learning representations of nuclei in histopathology images with contrastive loss." Medical Imaging with Deep Learning. PMLR, 2021.
> > >
> > > [6] Wang, Jingwen, et al. "Weakly supervised prostate tma classification via graph convolutional networks." 2020 IEEE 17th International Symposium on Biomedical Imaging (ISBI). IEEE, 2020.
> > >
> > > [7] Lee, Kibok, et al. "i-mix: A domain-agnostic strategy for contrastive representation learning." arXiv preprint arXiv:2010.08887 (2020).
> > >
> > > [8] Diao, James A., et al. "Human-interpretable image features derived from densely mapped cancer pathology slides predict diverse molecular phenotypes." Nature communications 12.1 (2021): 1-15.
> > >
> > > [9] Saltz, Joel, et al. "Spatial organization and molecular correlation of tumor-infiltrating lymphocytes using deep learning on pathology images." Cell reports 23.1 (2018): 181-193.
> > >
> > > [10] Gao, Guangshuai, et al. "Cnn-based density estimation and crowd counting: A survey." arXiv preprint arXiv:2003.12783 (2020).

---

> > > ### Comment · Reviewer_nwTd · 2022-11-29
> > > **Q4 and Q5**
> > >
> > > Fine with reply to Q4.
> > >
> > > As for reply to Q5, what do you mean by "what do you mean by "no spatial constraint exists over matching the distribution of representations across the views"

---

> > > > ### Author Response · Authors · 2022-11-30
> > > > **Response to Reviewer 2's (nwTd) question**
> > > >
> > > > We are thankful to the reviewer 2 (nwTd) for the feedback. We are happy to be able to satisfactorily address most of the concerns raised.
> > > >
> > > > Clarification for Q5 :- In the vanilla SSL frameworks for vision transformers, following the encoder backbone, “average pooling” is applied  before the projection head. Consequently, instead of matching domain-specific representations (cellular region and non-cellular regions in our case), the average pooling could cause the model to collapse. The model could just focus on a particular region (i.e. some prominent patterns) and optimize the SSL objective, without the need to focus on other diverse spatial locations.

---

> > > > > ### Author Response · Authors · 2022-12-12
> > > > > **Response to Reviewer 2 (nwTd)**
> > > > >
> > > > > Dear Reviewer nwTd,
> > > > >
> > > > > We hope our discussions clarify Q5. Kindly let us know if there is anything you would like us to clarify further. We would also like to point out that we have addressed Q6 in our first response part 3/3. We look forward to your further feedback and discussion.

---

> > ### Comment · Reviewer_nwTd · 2022-11-29
> > **Satisfied with replies to Q1, Q2 and Q3**
> >
> > Thanks for the responses to the first three questions, I'm now satisfied with the first 3 questions.

---

### Official Review · Reviewer_vgEj · 2022-10-25

**Confidence:** 4
**Correctness:** 3
**Technical Novelty And Significance:** 3
**Empirical Novelty And Significance:** 3
**Recommendation:** 6

**Clarity, Quality, Novelty And Reproducibility:**

The reviewer believes the idea of the disentanglement for the divergence of attention is novel but the role of the labeled images used in the training of cell nuclear segmentation is not small. In addition, the analysis of the resultant attention is not satisfactory. It would be better to compare the proposed method with multi-scale transformers in the introduction to clarify the strength of the proposed method. Existing methods that represent WSIs with multi-scale transformers aim to represent not only local features at the level of individual cells but also global features between cells and the cytoplasm. Attention matrices of more distant tokens are computed for the representation of the latter global features and would have some relationships with the diversified attentions obtained by the proposed method.

**Strength And Weaknesses:**

Strength:
As the authors mention in the paper, WSIs have characteristics different from natural images. In pathology images, cell nuclei are spatially dispersed, and the features of each cell nucleus, the spatial patterns of the multiple cell nuclei and the cytoplasm textures can all contribute to pathological image diagnosis. The authors point out the importance of the divergence of the attentions and find that the proposed method can diversify the attentions. The diversified attention would describe not only the relationships between spatially close patches but also the relationships between more distant patches distributed in wider areas. The experimental results demonstrate that the proposed method improves the performance of classification significantly.

Weakness:
The method employs a cell segmentation method that requires labeled data and hence the reviewer questions whether the proposed method can be classified as a self-supervised learning (SSL) method and whether the performance comparison with SSL methods is fair. The labeled data used in the training of the cell segmentation could implicitly teach that the nuclear regions would play important roles in the classification tasks.

The analysis of the obtained attentions is not satisfactory. The experimental results show that the attentions are diversified but the details of the obtained attentions between tokens are not indicated. As shown in the left image in Fig.5, each crop image would include pathologically different regions such as the necrosis regions and the tumor ones. The attention matrix can represent the relationships between the patch images. Did the attention matrices obtained by the proposed method and by other conventional methods represent the pathological difference of the regions? By disentangling the representation, did the representations obtained by the proposed method become more valid from a pathological point of view?


**Summary Of The Paper:**

This paper proposes a digital pathology representation learning method that uses a vision transformer. The method splits a given whole slide image (WSI) into multiple crops and the cropped images are represented by using multi-head self-attentions (MSAs): Each cropped image is split into patch images, the patches are encoded by MSAs, and an average pooling is applied to the feature vectors obtained by the MSAs as a feature vector of the crop. Different from conventional vision transformer, the proposed method separates the patches into two regions: A cell-region and a back-region. The former is a set of the patches with cell nuclei inside and the latter is a set of the patches without cell nuclei inside. The method computes six different feature vectors for each crop. Mixing the patches from both regions and using vision transformers, the proposed method obtains a feature vector for each of the two regions by applying average pooling to the resultant feature vectors of the patch images. Separating the patches for each region and applying vision transformer, the method obtains another feature vector for each region. This feature vector is obtained based on the context within each region. Other two feature vectors are obtained based on the context between the two regions, which is described by the attention matrix computed basically from pairs of two patches from different regions. It is shown that separating (disentangling) the patches into the two regions, one can diversify the strength of the attentions and the performance of the classifiers can be improved.


**Summary Of The Review:**

Because of the weakness described above, the reviewer rates this paper just below the borderline. I am willing to rase the rating if it is appropriately explained the reason why the proposed method can be classified as a SSL method.

---

> ### Author Response · Authors · 2022-11-09
> **Response to Reviewer 1's critique - [part 1/3]**
>
> We thank Reviewer 1 for their constructive reviews and suggestions. Here, we specifically respond to Reviewer 1’s critiques. We will update the paper and appendix, as appropriate.
>
> Q1) The method employs a cell segmentation method that requires labeled data and hence the reviewer questions whether the proposed method can be classified as a self-supervised learning (SSL) method.
>
> - We thank the reviewer for this question - to avoid any confusion, we are clarifying our choice to categorize our method as SSL.  Our method ensures that, for any new cancer type, we do not need expert annotations of cell segmentation. Instead, pretrained open-source segmentation tools like HoVer-Net can be used during data curation. These robust segmentation tools are trained on > 200K annotated cells, and due to their effective generalizability, they do not need  to be fine-tuned before performing downstream analysis [1, 2]. This is, in essence, a preprocessing step essential for data curation. We therefore  believe that our pre-training is indeed self-supervised. However if the reviewers and the area chair strongly believe that the provenance of the pre-existing tools should be taken into account, then we are happy to term it as a new very weakly-supervised representation learning or prior-guided representation learning strategy and modify the content in the paper accordingly. Note that none of the supervision necessary for those pre-existing tools is related to the problem that we are solving. Instead of engaging in philosophical arguments about the exact semantics of the term “self-supervision”, we can rephrase ‘SSL’ to any terminology that the reviewers and area chair consider more appropriate.
>
> - We believe that a terminology modification does not change the implementation or the fundamentals of our proposed contribution. Applying our method to a new digital pathology cancer dataset does not require any additional supervision. We emphasize that our novelty lies in the observations we present about the sparsity of attention in vanilla SSL models, its important implication in digital pathology, and our diversification of attention through dense matching in SSL pretext tasks when appropriate for the application domain.
>
> Q2)  Is the performance comparison with SSL methods fair? The labeled data used in the training of the cell segmentation could implicitly teach that the nuclear regions would play important roles in the classification tasks.
>
> - We agree with the observation that nuclear regions may guide classification, albeit implicitly. However, this does not render the comparisons unfair. In fact the premise of our work is built upon our observation of sparse attention for existing self-supervised representation learning techniques. Our goal is to effectively use domain knowledge during self-supervision to address this limitation. We would like to point out that for WSI-level classification tasks, localized annotations are often not available. Therefore the only potential way to pre-train the neural networks to encode rich representations is through self-supervision; they are the only baselines available. Hence our performance comparison is fair.
>
>
> [1] Lu, Wenqi, et al. "SlideGraph+: Whole Slide Image Level Graphs to Predict HER2 Status in Breast Cancer." Medical Image Analysis (2022): 102486.
>
> [2] Pati, Pushpak, et al. "Hierarchical graph representations in digital pathology." Medical image analysis 75 (2022): 102264.

---

> > ### Author Response · Authors · 2022-11-09
> > **Response to Reviewer 1's critique - [part 2/3]**
> >
> > Q3) ​​It would be better to compare the proposed method with multi-scale transformers in the introduction to clarify the strength of the proposed method. Existing methods that represent WSIs with multi-scale transformers aim to represent not only local features at the level of individual cells but also global features between cells and the cytoplasm.
> >
> > - We believe our work is not an alternative approach but rather a  complementary one to the recent multi-scale transformer, HIPT [3].  Our hypothesis is that the sparse attention by models pre-trained using vanilla SSL is especially detrimental for encoding representations, as the model would be sparsely encoding the information available in WSI-crops. To address this issue, we presented a dense-matching task to enforce the model to pay close attention to multiple regions in the crops. Incorporating our pre-training strategy in hierarchically training the multi-scale transformer HIPT could potentially further boost the performance of HIPT. While HIPT focuses on global-local representation learning, our method can complement this by diversifying attention in each component (global/local) module. This will be explored in our future work. As suggested by the reviewer, we will briefly mention this in the introduction of the paper. We will upload the revised paper before the end of the discussion period.
> >
> >
> > Q4) The analysis of the obtained attention is not satisfactory. The experimental results show that the attentions are diversified but the details of the obtained attention between tokens are not indicated.
> >
> > - The goal of our attention analysis is to quantitatively (please refer to Fig. 4 and Fig. 9 in the paper) and qualitatively (please refer to Fig. 5 and Fig. 10 in the paper)  show that our model de-sparsifies the attention, which we hypothesize would boost the encoding capabilities. However to address the reviewer’s concern regarding the interaction relationship between the tokens in the images, we have added an additional visualization figure in the appendix (section A.6 and Fig. 11 Tokens Interaction Visualization.). We illustrated how a given query (marked with a yellow bounding box) interacts with (attends to) other tokens by visualizing the row of an attention matrix corresponding to the query. Our findings in query-level interaction analysis is consistent with that of  previously shown WSI-crop level attention analysis, i.e, the query token sparsely attends to different tokens in baseline (vanilla SSL). In contrast, our pretraining alleviates this problem by performing a dense interaction of the query token with other tokens in the image.
> >
> > [3] Chen, Richard J., et al. "Scaling Vision Transformers to Gigapixel Images via Hierarchical Self-Supervised Learning." Proceedings of the IEEE/CVF Conference on Computer Vision and Pattern Recognition. 2022.

---

> > > ### Author Response · Authors · 2022-11-09
> > > **Response to Reviewer 1's critique - [part 3/3]**
> > >
> > > Q5) As shown in the left image in Fig.5, each crop image would include pathologically different regions such as the necrosis regions and the tumor ones. The attention matrix can represent the relationships between the patch images. Did the attention matrices obtained by the proposed method and by other conventional methods represent the pathological difference of the regions?
> > >
> > > - No, they do not represent the pathological difference of the regions for either method. In fact, identifying pathological differences via  this technique was never our intention; the goal was to ensure that the model takes into account the diverse information available through de-sparsification of attention. We however note that quantifying attention variability wrt different pathologies may be an interesting direction to pursue in the future.
> > >
> > > Q6) By disentangling the representation, did the representations obtained by the proposed method become more valid from a pathological point of view?
> > >
> > > - Yes, they are. The interaction between various entities (nuclei, stroma, glands, etc.) in pathology has been found to have clinical significance [4, 5, 6]. In this study, we aim to model the interaction between the cellular (comprising various types of cells) and non-cellular regions (comprising stroma, smooth muscle region, fat, etc). For modeling this interaction, we believe that disentangling both the regions followed by explicitly encoding the inter-intra region interaction is necessary. Otherwise  without disentanglement, encoding the inter-intra region interaction would not be precisely achievable, because average pooling in vision transformers could potentially dilute these token-level crucial signals. Future directions could delve into utilizing more refined entities (such as immune cells region, tumor regions, glands, necrotic region, and stroma) and quantifying their mutual interactions, thus better guiding the neural network to learn intricacies in digital pathology. We have uploaded a new version of the appendix and discuss this in section A.7.
> > >
> > > [4] Zormpas-Petridis, Konstantinos, et al. "Superhistopath: a deep learning pipeline for mapping tumor heterogeneity on low-resolution whole-slide digital histopathology images." Frontiers in oncology 10 (2021): 586292.
> > >
> > > [5] Diao, James A., et al. "Human-interpretable image features derived from densely mapped cancer pathology slides predict diverse molecular phenotypes." Nature communications 12.1 (2021): 1-15.
> > >
> > > [6] Saltz, Joel, et al. "Spatial organization and molecular correlation of tumor-infiltrating lymphocytes using deep learning on pathology images." Cell reports 23.1 (2018): 181-193.

---

> > ### Comment · Reviewer_vgEj · 2022-11-10
> > **Response to the authors' comments**
> >
> > Thank you for the response.
> >
> > The reviewer understands that the proposed method can learn without using labels of localized cancer regions and that it is one of the strength of the method. However, the knowledge that cell nuclei should be the focus in pathological images can be very powerful in cancer recognition. Cancer subtypes can often be identified based on the morphology of the cell nuclei and/or the spatial distribution of cells.
> >
> > In the proposed method, the knowledge that the cell nucleus should be focused on is given externally rather than being acquired through self-supervised learning, and the source of the knowledge is the large amount of annotated data used when learning HoVer-Net. Comparing with the use of super-pixels for image representation, the reviewer thinks we can understand the important role of HoVer-Net.
> >
> > The proposed method works well because the cell nuclei can be segmented accurately and hence the good performance of the proposed method seems brought about (at least partly) by a large number of the labeled data. The reviewer still strongly believes that the provenance of the domain knowledge introduced by HoVer-Net should be taken into account and would like to entrust the judgement of the validity of the comparison with SSL to the area chair.

---

> > > ### Author Response · Authors · 2022-11-11
> > > **Response to Reviewer 1's comment on 11/10**
> > >
> > > We thank the reviewer for acknowledging our method’s ability to learn without using the labels of localized cancer regions as one of its core strengths. We have also previously addressed the other critiques raised by Reviewer 1  (in part 2 and part 3 of our response on 11/9) and highlighted the other major contributions. We understand the reviewer’s perspective on taking the provenance of domain knowledge into account. However we would like to respectfully request the reviewer to provide us pointers regarding appropriate comparison studies to strengthen our experimentation setup. The sole reason we compared our method with vanilla-SSL is because of the lack of localized annotations available at WSI-crop level. Therefore, it would be helpful and constructive if the reviewer could point us to other specific methods in digital pathology to compare our approaches against. We will make every effort to conduct these additional comparisons before the end of the discussion period (11/18).

---

> > > > ### Comment · Reviewer_vgEj · 2022-11-11
> > > > **Response to the authors' comments.**
> > > >
> > > > Thank you.
> > > >
> > > > After reading the comments from the authors, I have come to believe that comparative experiments are indeed not easy. I write honestly (as I recall) what I thought when I first read your paper.
> > > >
> > > > When I started reading your paper, I focused on the word "SSL" , so I started reading the paper expecting it to be a method for acquiring knowledge about the structures of image (specific to the domain) over a large number of unannotated images. However, in section 3, it is described as the method uses domain knowledge as a prior (by cell-nucleus segmentation), which made me a little bit confused. There are many pathological image processing methods based on cell nucleus segmentation, such as the methods using cell-graph, as cited in section 3.2. And I felt that it should be obvious that focusing on cell nuclei improves performance.
> > > >
> > > > However, I have to say that it is not of course obvious how to incorporate prior knowledge.
> > > >
> > > > I am still not sure that we can say the propose method can be classified as SSL but am now willing to raising the grade.

---

> > > > > ### Author Response · Authors · 2022-11-15
> > > > > **Response to Reviewer 1’s comment (from Nov 11):-**
> > > > >
> > > > > - We sincerely thank the reviewer for showing willingness to raise the rating for our work. We have considered R1’s and AC’s suggestions and made changes accordingly (please see author response under :- “Terminology change: Di-SSL (Diversity inducing Self-Supervised Learning) changed to DiRL (Diversity inducing Representation Learning)”.)

---

> > > ### Comment · Area_Chair_9Y2o · 2022-11-13
> > > **Self-supervised method shall not rely much on annotated data for a similar task**
> > >
> > > Based on my understanding, whether a method can be considered as self-supervised or not depends on whether the method replying on annotated data. For this paper, I would say it depends on how the prior knowledge of cell segmentation is obtained and how it helps. If the cell segmentation is trained by a large dataset with annotated ground truth, and the accuracy of this  segmentation is also very important to the final improvements, I felt that it is inappropriate to say this is self-supervised method. Having said that, if the cell segmentation prior is done by hand-crafted approach without training from large labelled data, it is ok to say this is self-supervised as the approach does not really rely much on the manual labels.

---

> > > > ### Author Response · Authors · 2022-11-15
> > > > **Response to Reviewer 1’s (from Nov 11) and Area Chair’s (from Nov 13) feedback regarding SSL :-**
> > > >
> > > > - We thank the AC for their constructive insights regarding the appropriateness of the name of our method. We have thoroughly thought whether to term our method  SSL or a pre-training strategy. We believe our contribution is segmentation method-agnostic; it can still achieve de-sparsification with handcrafted approaches. Preliminary analysis (Figure 7 in appendix) seems to suggest that ViT pre-trained with prior from HistomicTK (handcrafted approach) may still be able to de-sparsify the attention. However, in order to avoid further confusion, we have modified the paper based on the R1’s and AC's  suggestions, since our method does rely on pre-trained segmentation tools. We would like to point the R1 and AC to the author's response under :- “Terminology change: Di-SSL (Diversity inducing Self-Supervised Learning) changed to DiRL (Diversity inducing Representation Learning)”. Consequently, in the revised text, instead of calling our method SSL, we modified it to “a diversity inducing or prior-guided pre-training strategy”.

---

### Author Response · Authors · 2022-11-15
**Terminology change: Di-SSL (Diversity inducing Self-Supervised Learning) changed to DiRL (Diversity inducing Representation Learning)**

Note: We have changed the name of our method from Di-SSL (Diversity inducing Self-Supervised Learning) to DiRL (Diversity inducing Representation Learning) based on recommendations of Reviewer 1 and Area Chair. This change has been reflected in the updated paper as appropriate. Also in the revised text, instead of calling our method SSL, we modified it to “a diversity inducing or prior-guided pre-training strategy”. This does not change our contributions, takeaways, or validity of our comparison studies.

Currently, the replacement is done only in PDF with the paper, since OpenReview does not allow changes to the abstract on the page with the paper.

---

### Decision · Program_Chairs · 2023-01-20

**Decision:**

Reject

**Justification For Why Not Higher Score:**

See my the summary and the discussion between AC and reviewers.

**Justification For Why Not Lower Score:**

N/A

**Metareview: Summary, Strengths And Weaknesses:**

The main claim from the authors is that self-supervised learning cause sparse attention maps, instead of data. This paper proposed an approach to use cell-segmentation as prior to make the attention maps less sparse and improve the results.

Despite some argument whether the approach by using cell-segmentation trained from large amount of labelled data can be considered as self-supervised or not, the main claim from the authors lack of experimental support. The claim is: "The model tends to localize most of its attention to a small fraction of regions, leading to sub-optimal representation learning. To further validate our observation, we visualized the attention maps of a self-supervised ImageNet pre-trained model on natural images (see Fig. 1). Similar observations led us to conclude that this is a property of SSL rather than of data.”   Based on my understanding, simply looking at the attention maps from SSL trained models alone is not strong enough to proof that this claim is true.  The authors need to design a proper experimental validation to justify this, for example, by comparing the attention maps with SSL against those without SSL.

In addition, there is a consensus that the algorithm is very specific and requires segmentation model trained from a large amount of data, which makes it not generic for other applications.

**Summary Of Ac-Reviewer Meeting:**

1) During the discussion between AC and reviewers, we cannot agree with the claim that “this is a property of SSL rather than of data”.
2) We agree that the improvement comes from cell segmentation prior instead of desparsificiation.
3) There is a consensus that the approach proposed here is not generic and it relies on high accurate segmentation prior, which makes it challenge to generalize for other applications.